# Nanoparticles exhibiting virus-mimic surface topology for enhanced oral delivery

Zhentao Sang[1,2,8], Lu Xu[3,8], Renyu Ding[4,8], Minjun Wang[1], Xiaoran Yang[1], Xitan Li[4], Bingxin Zhou[1], Kaijun Gou[5], Yang Han[6], Tingting Liu[7], Xuchun Chen[7], Ying Cheng [7]✉, Huazhe Yang [2]✉ & Heran Li [1]✉

The oral delivery of nano-drug delivery systems (Nano-DDS) remains a challenge. Taking inspirations from viruses, here we construct core–shell mesoporous silica nanoparticles (NPs, ~80 nm) with virus-like nanospikes (VSN) to simulate viral morphology, and further modified VSN with L-alanine (CVSN) to enable chiral recognition for functional bionics. By comparing with the solid silica NPs, mesoporous silica NPs and VSN, we demonstrate the delivery advantages of CVSN on overcoming intestinal sequential barriers in both animals and human via multiple biological processes. Subsequently, we encapsulate indomethacin (IMC) into the nanopores of NPs to mimic gene package, wherein the payloads are isolated from bio-environments and exist in an amorphous form to increase their stability and solubility, while the chiral nanospikes multi-sited anchor and chiral recognize on the intestinal mucosa to enhance the penetrability and ultimately improve the oral adsorption of IMC. Encouragingly, we also prove the versatility of CVSN as oral Nano-DDS.

In recent years, nano-drug delivery systems (Nano-DDS) with a large spectrum of composition[1–3], morphology[4–6], and multi-functional inner/outer surface designs[7,8] have been developed and shown potential research values in the field of oral drug delivery. Other than the size effect, some engineered Nano-DDS also provide smart stimulus-triggered[9] and/or both passive and active targeted drug release performance to achieve satisfactory delivery efficiency and reduce side effects[10]. Nevertheless, synthetic Nano-DDS still faces the challenges of effective transporting through the intestinal mucosa and entering into the blood circulation[11]. One of the primary reasons for the invalidation of oral Nano-DDS is the weak adhesion between the nanoparticles (NPs) and the bio-interfaces (including the intestinal mucosa and the cell membrane), and the subsequent inadequate uptake of NPs by the epithelial cells[12]. To the best of our knowledge, Nano-DDS are conventionally designed as spheres, as nanospheres are spontaneously formed to maintain the hydrostatic equilibrium and minimize the surface free energy, and meanwhile show good stability, high symmetry and favorable mobility in bio-systems[13,14]. Despite these potential advantages, spheres always mean the smallest surface areas at constant volumes and inadequate contact areas (single point contact) on the basis of geometry, which directly influence their biological behaviors[15], resulting in short retention times and quick elimination by air/fluid flows. These factors ultimately limit the delivery efficiency of spherical Nano-DDS. In other words, the morphology and surface topology of NPs deeply affect their oral adsorption performance[16].

Viruses, which are nanosized entities composed of proteins and nucleic acids, have emerged as prospective nanocarriers to transport the cargos across various barriers in the body, and have become valued inspiration sources for the design of novel Nano-DDS[17,18]. For example, adenoviruses (ADV) and adeno-associated viruses (AAV) have a long history of clinical development and have gradually emerged as ideal gene delivery platforms for a widely range of diseases[19,20], such as

[1]School of Pharmacy, China Medical University, Shenyang 110122, China. [2]School of Intelligent Medicine, China Medical University, Shenyang 110122, China. [3]School of Pharmacy, Shenyang Pharmaceutical University, Shenyang 110016, China. [4]Department of Intensive Care Unit, The First Hospital of China Medical University, Shenyang 110001, China. [5]Department of Pathology, The First Hospital of China Medical University, Shenyang 110001, China. [6]Institute of Tibetan Plateau, Southwest Minzu University, Chengdu 610225, China. [7]Department of Organ Transplantation and Hepatobiliary, The First Hospital of China Medical University, Shenyang 110001, China. [8]These authors contributed equally: Zhentao Sang, Lu Xu, Renyu Ding. ✉e-mail: chengying75@sina.com; hzyang@cmu.edu.cn; liheranmm@163.com

cancer[21,22], acquired immune deficiency syndrome (AIDS)[23,24], and Alzheimer's disease (AD)[25,26]. Other than the direct application of viral NPs (VNP) for gene therapy, Lu et al. utilized hepatitis B core protein (HBc) virus like particles (VLPs) as delivery platform of a lipophilic near infrared dye IR780 to resemble the package of capsid protein and viral core, and substantially improved the aqueous and photostability of IR780[27]. Jia et al. constructed peptidyl VLPs by mimicking the structural function of the human immunodeficiency virus (HIV), encapsulated the DNA of clustered regularly interspaced short palindromic repeat associated proteins 9 (CRISPR/Cas9), and demonstrated that peptidyl VLPs could penetrate the cellular membrane through the viral entry route and targeted delivery Cas9 gene into the cell[28]. Inspired by the rabies virus (RABV), Youn's group developed silica coated gold nanorods (RVG-PEG-AuNRs@SiO$_2$) for the treatment of brain gliomas, wherein the shape (bullet-like) and size (-120 × 150 nm) of gold nanorods (AuNRs) was similar to the appearance of the RABV, and the surface-modified functional glycoprotein (RVG29) specifically interacted with the nicotinic acetylcholine receptor (AchR) expressed on neuronal cells to enter the central nervous systems and deliver the therapeutic agent[29]. Apart from the nanoscale size and functional capsid protein, the intrinsic surface topologies of viruses, referred to the shape, curvature, texture and roughness of the surface, and the density and spatial orientation of the functional groups, have also been recognized as the primary determinants for their invasion competence[30,31]. To be specific, the external nanospikes on viruses provide abundant binding sites and directly anchor onto the biomembranes, while their heterogeneous rough surface significantly increases the frictional force and reduces the repulsive force, thus facilitating the endocytosis or signal transduction[32]. Moreover, as natural creations, all viruses exhibit chiral architectures, wherein the helical structure of DNA or RNA in the core plays a vital role in the gene transcript, translation and expression of viruses, while the secondary structure of proteins is crucial to maintain their spatial stability and bio-responsive activity[33,34].

In this work, taking inspirations from the extremely high infectivity and penetrability of viruses to tissues and cells, we construct an efficient oral Nano-DDS by mimicking both the structure and function of viruses. We first prepare mesoporous silica NPs with nanoscale virus-like spikes (VSN) via an epitaxial growth strategy to simulate the morphology of viruses and improve the surface/interface roughness. Considering that the biological functions of natural assemblies are highly dependent on their chirality, we modify VSN with L-alanine (L-Ala), denoted as CVSN, to enable chiral recognition. During the delivery task, the core–shell structures of CVSN resemble the loading of genes and isolate the payloads from the bio-environment to improve their stability, while the nanospikes on the surface multi-sited anchor on the bio-membranes and chiral recognize with the membrane protein for easier engulfment by cells. As a proof-of-concept, the structure, physicochemical and surface/interface properties, as well as the biosafety of CVSN are investigated by comparing with those of smooth solid silica NPs (SSN), spherical mesoporous silica NPs (MSN) and VSN. Then, the oral delivery capacities and mechanisms of NPs, including their bio-adhesion, mucus penetrability, intestinal permeability, GI tract retention, oral adsorption and cellular uptake, are systematically evaluated from the perspective of functional bionics. We then encapsulate insoluble nonsteroidal anti-inflammatory drug (NSAID) indomethacin (IMC)[35,36] into the nanopores of NPs to simulate the gene package and assess the drug loading and release manners in vitro and the pharmacokinetic and pharmacodynamic properties in vivo. Notably, we also choose a series of NSAIDs classified as biopharmaceutical classification systems (BCS) 2–4 characterized by poor solubility and/or low permeability to investigate the versatility of CVSN as oral Nano-DDS. It is expected that, by mimicking the shape, size, surface topology and chiral recognition process of viruses, CVSN can be employed for efficiently overcoming the multiple physiological barriers in the GI tract and achieve favorable oral drug delivery efficiency.

## Results
### Morphology and structure of NPs

To simulate the structure and function of viruses, we first prepared virus-like mesoporous silica NPs exhibiting spiky rough surface (VSN) via a single-micelle epitaxial growth in a biphase reaction system, with a low surfactant concentration, which allowed the coassembly of reactants at the oil–water interface (for realizing continuous interfacial growth; Fig. 1a)[37]. Next, we modified the chiral group L-Ala onto VSN via two-step reactions, including amination (AVSN) and acylation to endow chirality (CVSN) for functional bionics (Supplementary Fig. 1). In this term, NPs with different surface topology, including SSN (smooth surface), MSN (porous surface), VSN (spiky rough surface) and CVSN (chirality and spiky rough surface) were synthesized. All the FTIR spectra of NPs showed broad peaks of v–Si-O-Si, v–Si-OH and δ–Si-O-Si at 3450, 1076 and 462 cm⁻¹, respectively (Fig. 1b). The FTIR spectra of AVSN presented a peak of v–N-H at 1633 cm⁻¹, confirming the amino modification, which was in accordance with the X-ray photoelectron spectroscopy (XPS) finding of element N (Supplementary Fig. 2). After the grafting of L-Ala, the FTIR spectra and the XPS spectra of CVSN showed new bands assigned to carbonyl group at 1694 cm⁻¹ and the C1s, respectively (Fig. 2a).

As presented in the TEM, SEM, and AFM images, SSN and MSN were uniformed nanospheres, while VSN AVSN and CVSN were virus-like particles with a large number of nanotubes distributed around their surfaces (Fig. 1c, d). All NPs were homogeneous in shape and size, with average particle sizes of 80 nm and hydrated particle sizes of 130 nm (Fig. 1e, f and Supplementary Fig. 2). The XPS spectra of SSN, MSN, and VSN demonstrated the existence of elements Si and O, while N were detected in AVSN and CVSN (Fig. 2e and Supplementary Fig. 2). Additionally, the weight loss (from 25 to 700 °C) of SSN, MSN, VSN, AVSN, and CVSN were 0.91%, 2.51%, 7.33%, 14.32%, and 22.07%, respectively, demonstrating the thermostability of silica framework and the successful grafting of functional groups (Fig. 2b). CVSN experienced intense weight loss on account of the organic chiral molecule modification, which was measured to be 14.75%. The presence of mesopores in MSN, VSN, AVSN, and CVSN was proven by small-angle X-ray scattering (SAXS) patterns (Fig. 2c and Supplementary Fig. 3). Besides, MSN, VSN, AVSN and CVSN all showed distinct type IV adsorption/desorption isotherms with hysteresis loops in the high-pressure section, most likely to be the pores formed by the stacking of NPs (Fig. 2d). By contrast, distinct isotherm was observed for SSN, demonstrating the absence of porous structure. The pore size distribution curve and the calculated texture parameters of NPs were summarized in Fig. 2e and Table 1, respectively, in which the pore sizes of MSN, VSN, and CVSN were 3.2, 2.9, and 2.5 nm, respectively.

### Surface/interface properties of NPs

First, the surface hydroxyl densities (D$_{SSH}$) of SSN, MSN, and VSN calculated from their weight losses and specific surface areas (S$_{BET}$) were 1.60, 4.65, and 5.57 (Si-OH/nm²), respectively (Table 1). The oil-water partition coefficients (P$_{ow}$) of NPs all showed negative values, indicating the hydrophilicity nature (Fig. 2f). As an essential indicator of the surface/interface properties, the wettability of NPs showed a clear gradient related to the surface roughness and surface hydroxyl density, wherein the initial contact angles followed the sequence of SSN > MSN > VSN > CVSN (Fig. 2g, h). Notably, compared to the spherical SSN and MSN, the contact angles of VSN and CVSN dropped to 0° in a very short time because their rough spiky surface topology provided multiple contact sites, resulting in high interfacial reactivity. For CVSN, the chiral modification further enhanced the roughness and wettability of the surface as evidenced by the reduced contact angle.

Moreover, the smallest pore size of CVSN generated the most potent capillary condensation and strongest additional pressure according to the Kelvin equation[38]. CD analysis was carried out to elucidate the chiral characteristics of the NPs, wherein CVSN (grafted with L-Ala) and DVSN (grafted with D-Ala for comparison) inherited the chirality of L/D-Ala, and showed positive and negative cotton effects in the range of 210–250 nm, respectively, indicating the successful chiral modification (Fig. 2i, j and Supplementary Fig. 3). In addition, NPs with different chiral configurations displayed distinct amino acid adsorption properties. To be specific, the CD spectra of L-Ala blue shifted to lower wavelengths and showed reduced peak area after incubation with CVSN exhibiting L-configuration, but red shifted after exposure to DVSN, indicating the variation on the interactions between NPs and chiral molecules and their aggregation states in the chiral microenvironment (Fig. 2k). A similar phenomenon was observed for D-Ala exhibiting opposite chirality, wherein the maximum absorption wavelength red and blue shifted after exposure to CVSN and DVSN, respectively. Moreover, corresponding symmetrical cotton effects were observed after the adsorption between L/D-Ala and CMSN/DVSN exhibiting different chirality. We then quantitatively analyzed the adsorption amount, and the results indicated that CVSN showed stronger adsorption capacity on L-Ala (17.58 µg/mg) than D-Ala (7.72 µg/mg), while DVSN showed a higher affinity on D-Ala than on L-Ala (22.47 and 8.22 µg/mg, respectively; Fig. 2l), implying the potential chiral recognition of CMSN/DVSN in the chiral microenvironment, and would beneficial for functional bionics on viruses.

Additionally, NPs were immersed in simulated gastric fluid (SGF, pH 1.2), simulated intestinal fluid (SIF, pH 6.8) and simulated body fluid

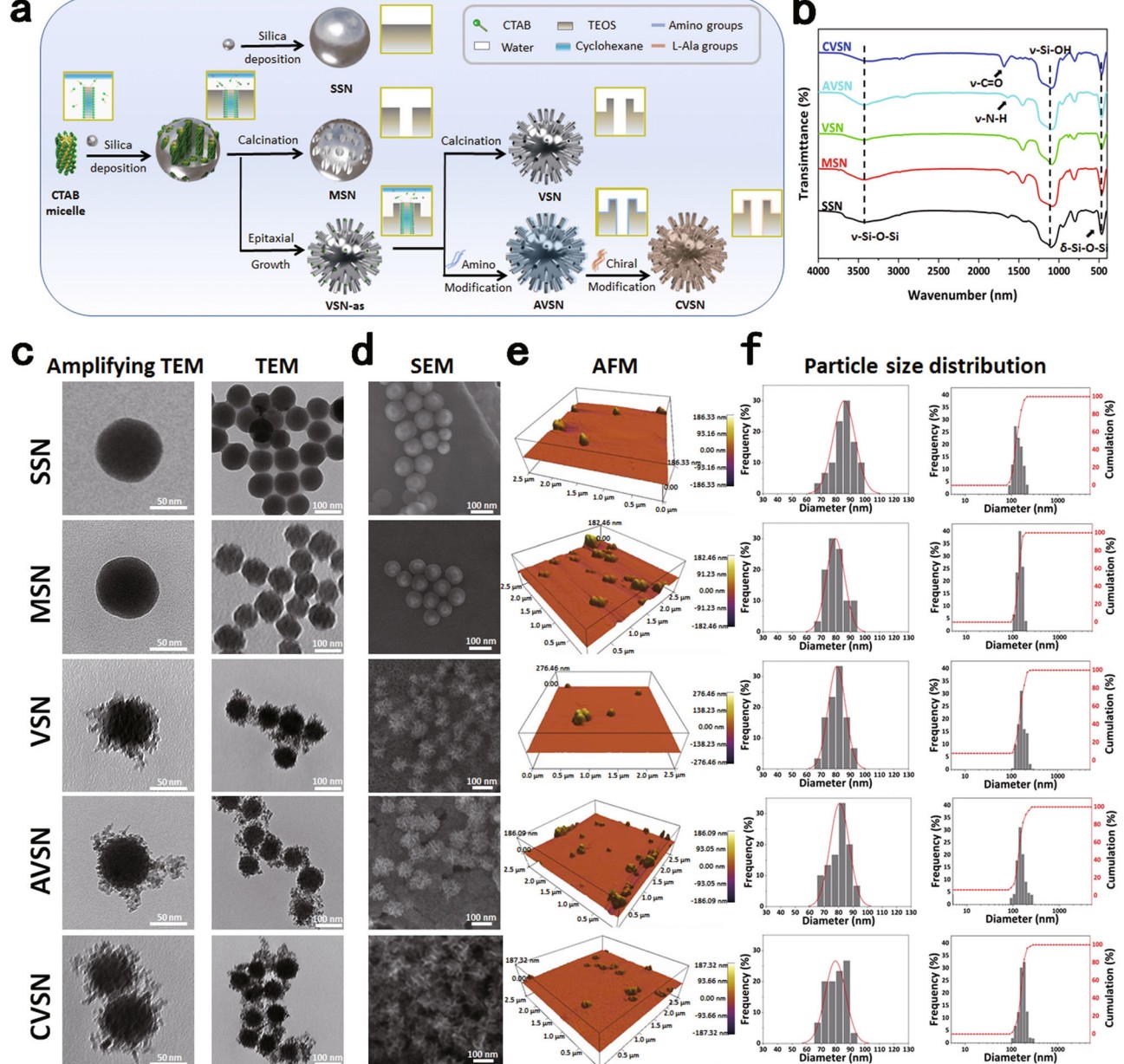

**Fig. 1 | The formation and morphology of NPs. a** Synthesis steps of NPs. **b** Fourier transform infrared (FTIR) spectra of NPs. Source data are provided as a Source Data file. **c** Transmission electron microscope (TEM) images of NPs. Experiment was repeated three times independently with similar results. **d** Scanning electron microscope (SEM) images of NPs. Experiment was repeated three times independently with similar results. **e** Atomic force microscope (AFM) images of NPs. **f** Particle size distribution (counted from the SEM images, and measured by dynamic light scattering [DLS], respectively) of NPs. Source data are provided as a Source Data file.

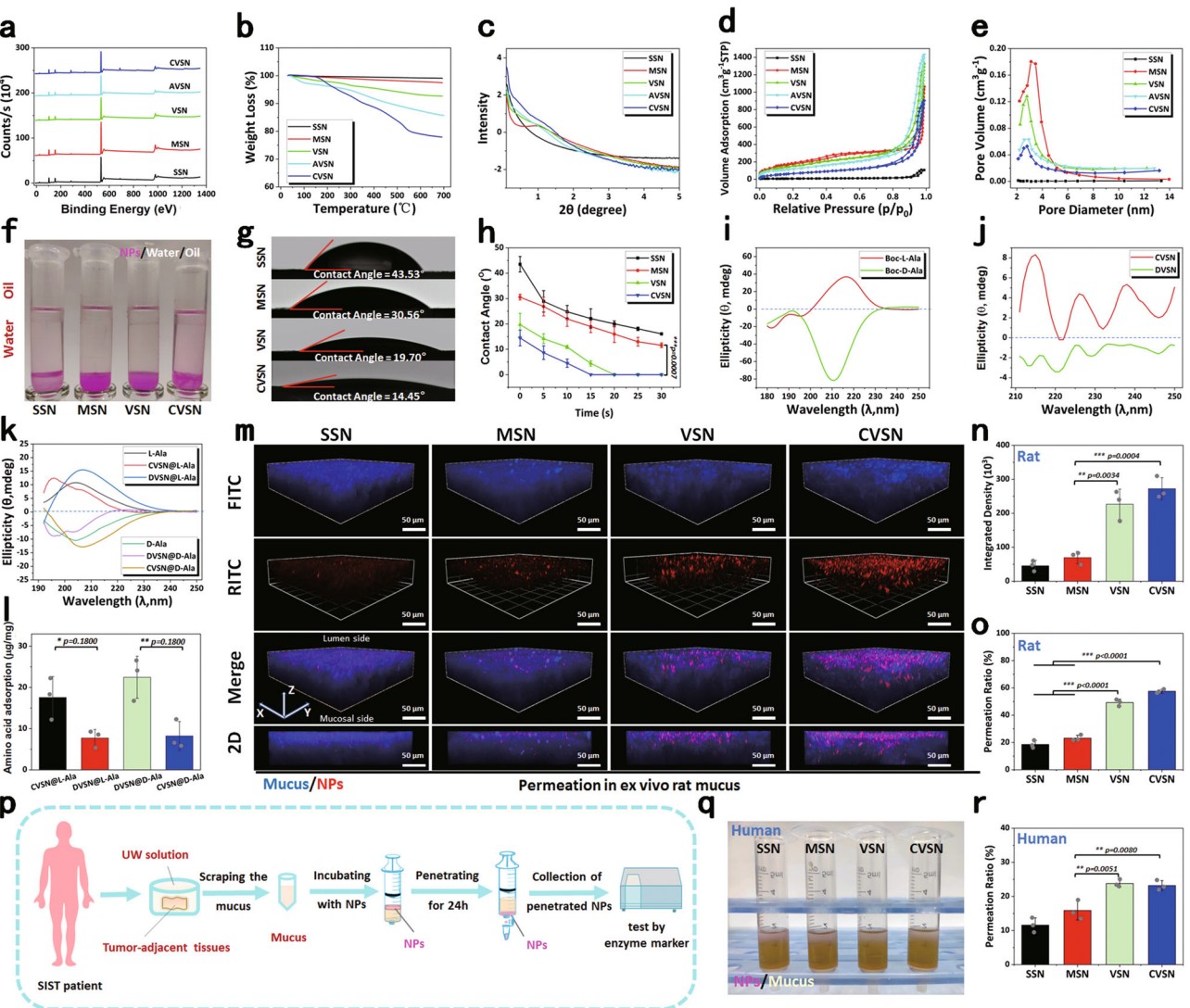

**Fig. 2 | The structure and surface properties of NPs. a** XPS spectra, **b** thermogravimetric analysis (TGA) curves, **c** Small angle X-ray scattering (SAXS) patterns **d** N₂ adsorption/desorption isotherms and **e** pore size distribution curves of NPs. Source data of **a**–**e** are provided as a Source Data file. **f** Oil-water distribution images of NPs. **g** Initial contact angles images of NPs. **h** The contact angles-time curves of NPs. Data are presented as the mean ± SD ($n = 3$ independent experiments). **$P < 0.01$ by two-tailed Student's $t$-test. Source data are provided as a Source Data file. Circular dichroism (CD) spectra of **i** Boc-L-alanine and Boc-D-alanine (Boc-L/D-Ala), **j** CVSN and DVSN, which was synthesized by grafting D-Ala onto the surface of VSN. Source data of **i**, **j** are provided as a Source Data file. **k** CD spectra of amino acid solutions before and after incubation with CVSN and DVSN. Source data are provided as a Source Data file. **l** Amino acid adsorption capacity of CVSN and DVSN. Data are presented as the mean ± SD ($n = 3$ independent experiments). *$P < 0.05$ and **$P < 0.01$ by two-tailed Student's $t$-test. Source data are provided as a Source Data file. **m** Three dimensions (3D) confocal fluorescence microscope (CLSM) images of NPs penetrated through the rat mucus layer; blue: mucus stained with fluorescein isothiocyanate (FITC). red: Rhodamine B isothiocyanate (RITC) labeled NPs. Scale bar: 50 μm. Depth: 70 μm. Experiment was repeated three times independently with similar results. **n** Fluorescence integrated density and **o** permeation ratio of NPs in mucus permeation. Data are presented as the mean ± SD ($n = 3$ independent experiments). **$P < 0.01$ and ***$P < 0.001$ by two-tailed Student's $t$-test. Source data of **n**, **o** are provided as a Source Data file. **p** Treatment protocols on the permeation of NPs in human intestinal mucus. **q** Image and **r** permeation ratio on the permeation of NPs in human intestinal mucus. Data are presented as the mean ± SD ($n = 3$ independent experiments). **$P < 0.01$ by two-tailed Student's $t$-test. Source data are provided as a Source Data file.

(SBF, pH 7.4) to evaluate their stability and degradation tendency via weight loss and SEM. In general, NPs showed controllable degradation manner in all mediums. Mild weight losses (5.50–13.67%) were observed for the short term incubation of NPs within 48 h, providing the basis for construction of Nano-DDS (Supplementary Figs. 4–7). For the long term, all NPs underwent biodegradation, bring less safety concerns (Supplementary Fig. 8). The degradation rates were affected by pH, and followed the order of SBF > SIF > SGF. In addition, for in intro and in vivo tracing, RITC was incorporated into the NPs through covalent bond, and showed good fluorescence stability (Supplementary Figs. 9–10).

Taken together, by mimicking the structure and function of viruses, CVSN showed active surface/interface properties, e.g., uniformed mesopores, abundant functional groups, multiple contact sites, high stability, favorable wettability and potential chiral recognition ability, which would be beneficial for its ex vivo and in vivo applications.

### Penetration of NPs through the intestinal mucus

Primarily, we explored the permeability of NPs through the viscoelastic intestinal mucus, which serves as one of the crucial barriers preventing the oral absorption of drugs and Nano-DDS[39–41]. The 3D CLSM images

## Table 1 | Texture properties of NPs

| Sample | $S_{BET}$ (m²/g) | $V_t$ (cm³/g) | $W_{BJH}$ (nm) | $D_{SSH}$ (-OH/nm²) |
|--------|------------------|---------------|----------------|---------------------|
| SSN    | 31.06            | 0.16          | /              | 1.60                |
| MSN    | 633.33           | 1.64          | 3.2            | 4.65                |
| VSN    | 544.28           | 2.05          | 2.9            | 5.57                |
| AVSN   | 435.71           | 2.21          | 2.6            | /                   |
| CVSN   | 256.86           | 1.40          | 2.5            | /                   |

showed that, only weak fluorescent signals from SSN and MSN were distributed in the mucus on the proximal luminal side of the intestine, whereas potent fluorescent signals from VSN and CVSN were observed in the entire mucus layer with good extensibility (Fig. 2m–o). Particularly, CVSN could effectively diffuse into the deepest and widest sections of the mucus layer, as evidenced by the co-localization of fluorescence signals in the 2D plane orientation images.

We then gathered the human intestinal mucus, and allowed NPs to permeate in the mucus (Fig. 2p). According to Fig. 2q, r, after 24 h of free diffusion, most SSN and MSN still remained on the top of the

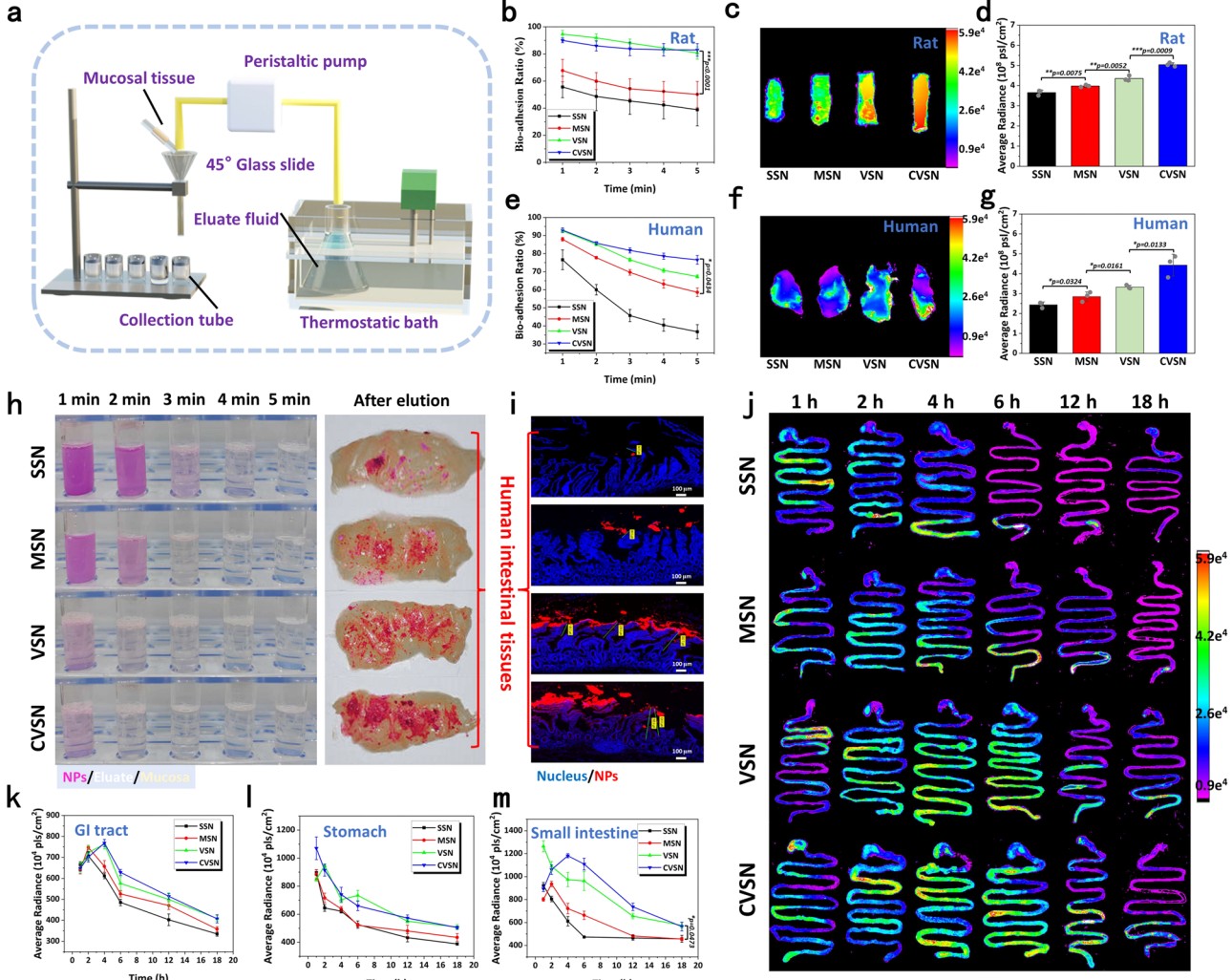

**Fig. 3 | Bio-adhesion ability of NPs on mucosal tissues. a** Schematic illustration of the apparatus on bio-adhesion study. **b** The bio-adhesion profiles of NPs on the rat intestinal mucosa measured by weight loss. Data are presented as the mean ± SD (n = 3 independent experiments). ***P < 0.001 by two-tailed Student's *t*-test. Source data are provided as a Source Data file. **c** Fluorescence images and **d** average intensity of NPs remained on the rat intestinal mucosa after elution. The color bar indicates the fluorescence intensity. Experiment was repeated three times independently with similar results. Data are presented as the mean ± SD (n = 3 independent experiments). **P < 0.01 and ***P < 0.001 by two-tailed Student's *t*-test. Source data are provided as a Source Data file. **e** The bio-adhesion profiles of NPs on the human intestinal mucosa measured by weight loss. Data are presented as the mean ± SD (n = 3 independent experiments). *P < 0.05 by two-tailed Student's *t*-test. Source data are provided as a Source Data file. **f** Fluorescence images and **g** average intensity of NPs remained on the human intestinal mucosa after elution. The color bar indicates the fluorescence intensity. Experiment was repeated three times

independently with similar results. Data are presented as the mean ± SD (n = 3 independent experiments). *P < 0.05 by two-tailed Student's *t*-test. Source data are provided as a Source Data file. **h** The bio-adhesion images of RITC labeled NPs performed on the human intestinal mucosa. Experiment was repeated three times independently with similar results. **i** CLSM images of RITC labeled NPs remained on the human intestinal mucosa; blue: nuclei of the intestinal mucosa stained with 4,6-diamidino-2-phenylindole (DAPI), red: RITC labeled NPs, scale bar: 100 μm. Experiment was repeated three times independently with similar results. **j** Fluorescence images of NPs remained on the GI tract after oral administration. Experiment was repeated three times independently with similar results. The color bar indicates the fluorescence intensity. Average intensity of NPs remained in the **k** GI tract, **l** stomach and **m** small intestine after oral administration. Data are presented as the mean ± SD (n = 3 independent experiments). *P < 0.05 by two-tailed Student's *t*-test. Source data are provided as a Source Data file.

mucus layer, while almost no VSN and CVSN were distributed on the top of the mucus. We then isolated the mucus from the bottom layer and measured the permeation rates of SSN, MSN, VSN and CVSN, which were 11.58%, 15.88%, 23.80%, and 23.19%, respectively. Similar results were observed for the permeation of NPs in rat intestinal mucus, with the permeation rates of 10.47%, 17.25%, 23.68%, and 27.01%, respectively (Supplementary Fig. 11).

## Bio-adhesion and retention of NPs on the intestinal mucosa

As a prerequisite for oral adsorption, we investigated the bio-adhesion manners of NPs on the intestinal mucosa tissues of rat and human via the elution method[42] (Fig. 3a). Before analysis, we first treated the intestinal tissues under the storage condition and the experimental procedure, and confirmed proper mucus coverage on the ex vivo mucosa (Supplementary Fig. 12). In both rat and human intestinal tissues, the bio-adhesion ability of NPs followed the order of CVSN > VSN > MSN > SSN, wherein the bio-adhesion rates were respectively 40.22%, 53.98%, 78.54% and 82.95% on the rat intestine, and 34.75%, 57.25%, 67.05%, and 77.45% on the human intestine (Fig. 3b–g). Spherical SSN and MSN with limited contact sites showed weak bio-adhesion followed by the quick elimination by the fluid flows, while virus-like VSN and CVSN presenting nanospikes on their surface multi-sited anchored on the intestinal mucosa, and showed higher bio-retention ability with significantly stronger fluorescence intensity detected on intestinal mucosa. Conformance to the surface/interface properties, CVSN exhibiting the marriage of virus-mimic spikey surface and molecular chirality could effectively adhere on the intestinal mucosa with both highest retention intensity and penetration depth (~398.4 μm) compared to the counterparts (Fig. 3h, i and Supplementary Fig. 13). The bio-adhesion ability of CVSN was also higher than that of DVSN with the non-dominant conformation, implying active chiral recognition with the membrane proteins (Supplementary Fig. 14). We also premixed the NPs labeled with different fluorescent dyes and performed the competitive elution procedure on human mucosa, wherein NPs with virus-like shape as well as L-chiral conformation always defeated the counterparts and showed higher retention (Supplementary Fig. 15). From this knowable, the virus-mimic strategy had considerably increased the interfacial reactivity, produced positive effects on mucosal adhesion via multi-sited anchoring and active chiral recognition.

To make the best use of Nano-DDS in the body, it was crucial to make sure whether the improved bio-adhesion of NPs would lead to the prolonged retention time in the GI tract and the subsequent satisfactory oral bioavailability. The retention behaviors of NPs were visualized using the IVIS Lumina Series III Living Image System. For SSN, strong signals were found in the small intestine at 1 h after oral administration, gradually weaken thereafter, and almost disappeared after 6 h (Fig. 3j). A similar phenomenon was observed in the case of MSN, wherein the fluorescent signals were only detected in the excretion sites (colorectal) instead of the absorption sites at 6 h post administration, and were basically excreted after 12 h. Corresponding to the bio-adhesion findings on the isolated intestinal segments, spherical SSN and MSN with the single contact site showed poor friction with the intestinal mucosa, resulting in short retention time and extremely fast excretion in the GI tract (Fig. 3k–m). In contrast, the fluorescence intensities of VSN and CVSN were significantly stronger in the small intestine at all time points, and the fluorescence signals were still detectable at 12 h after administration, indicating longer retention periods and higher probabilities for absorption.

## Absorption of NPs through the intestinal villi

To monitor the adsorption of NPs through the intestinal villi, animals were oral administered with RITC labeled SSN, MSN, VSN and CVSN,

and their middle small intestines were isolated and visualized (Fig. 4a). Unsurprisingly, the penetration and absorption of CVSN on the small intestine was most efficient due to their aforementioned virus-mimic surface topological advantages. To be specific, the fluorescence signals of SSN and MSN were extremely weak and mainly gathered in the center of the intestinal lumen rather than entering the villi, implying the poor bio-retention and mucosal penetration capacities (Fig. 4b–d). In contrast, VSN was noticed around the intestinal villi with higher fluorescence intensity than SSN and MSN. The transmembrane capacity of NPs was significantly enhanced after modification of L-Ala, wherein CVSN uniformly distributed along the surface of intestinal villi with the strongest intensity, which was also higher than that of DVSN exhibiting almost the same structural parameter but different molecular chirality (Supplementary Fig. 16). Besides, in the competitive adsorption of SSN vs VSN, SSN vs CVSN, MSN vs VSN and MSN vs CVSN, NPs with virus-like shape as well as L-chiral conformation always showed overwhelmingly superiority on oral adsorption according to the potent fluorescence signals (Fig. 4e, f and Supplementary Fig. 17). Bio-TEM images disclosed that SSN and MSN rarely entered the microvillis on the epithelial cells, however, a large number of VSN and CVSN could be observed around the microvillis (Fig. 4g, h). Moreover, CVSN could be internalized by the epithelial cells from duodenum, jejunum and ileum, and was mostly distributed in jejunum (Fig. 5a). Meanwhile, the bio-TEM images indicated no intercellular transport (Supplementary Fig. 18). We also investigated whether NPs could transport via intercellular space via immunofluorescence, wherein no significant down-regulation on the key proteins (including zonula occludens-1 [ZO-1] and occludin) associated with the functionality on the tight junctions of intestinal epithelial cells were found after incubated with NPs (Supplementary Fig. 18).

Taking into account the species variation, RITC labeled NPs were incubated with the ex vivo human and rat intestinal tubes by simulating the in vivo environment. Excitingly, CVSN still kept the best transmembrane capability with the strongest red fluorescence signal captured through the small intestinal villi (Fig. 5b–d and Supplementary Figs. 19–21). Besides, as is well-known, the ATP-driven drug-efflux pump p-glycoprotein (P-gp) would make notable impact on the oral adsorption of drugs and NPs, as it might lead to the efflux of the transported substrate[43,44]. We pre-incubated the intestinal tubes with the P-gp inhibitor verapamil (VER) and observed the uptake of NPs through the small intestinal villi. The results showed parallel absorption efficiency for VSN and CVSN before and after treatment with VER, but slightly increased fluorescence intensity for SSN after the incubation with VER, suggesting a P-gp efflux mechanism for the incompact uptake of smooth nanospheres (Supplementary Figs. 19–21 and Fig. 5e, f). It was apparent that, the surface topology of NPs was likely to be an important factor to determine their biological effects. In addition, by simulating the viral infection, the active interaction and retention of virus-like NPs may irritate or harm the mucosa on the GI tract. After exposure to SSN, MSN, VSN, and CVSN, no evident abnormalities, irritation, or damages were found on the intestinal mucosa of animals and human (Supplementary Fig. 22 and Fig. 5g). The immunofluorescence images of ZO-1 and occludin in turn suggested the integrality of tight junctions of intestinal epithelial cells and discovered no damage on the intestinal barrier (Supplementary Fig. 18).

Subsequently, we investigated the time-dependent adsorption of CVSN via bio-TEM and CLSM. The uptake of CVSN was started at 2 h post administration, and still be noticeable on the microvilli of epithelial cells at 24 h (Fig. 5h). Accordingly, CVSN was quickly adsorbed via small intestinal villi at 1 h post administration, highly aggregated around the intestinal villi and indeed transported to the basolateral (BL) side of intestinal cavity between 1 and 4 h. During the whole procedure, the uptake amount of CVSN was always higher than that of VSN (Fig. 5i–k and Supplementary Fig. 21).

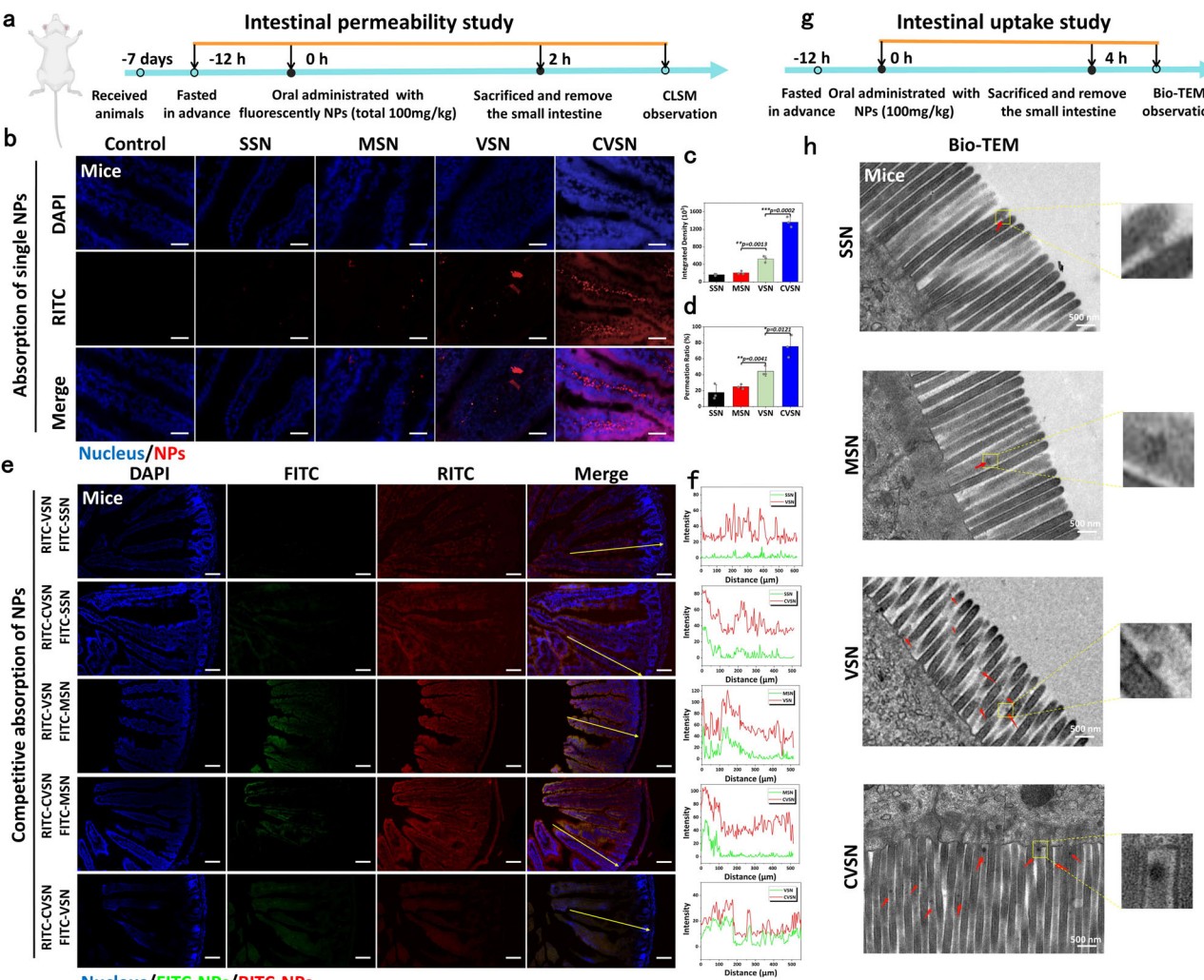

**Fig. 4 | Adsorption of NPs through the intestinal villi. a** Treatment protocols on the intestinal permeability study of NPs. **b** CLSM images of NPs absorbed at the small intestinal villi of mice at 2 h post oral administration; blue: nuclei of the intestinal villi stained with DAPI, red: RITC labeled NPs, scale bar: 50 μm. Experiment was repeated three times independently with similar results. **c** Fluorescence integrated density and **d** permeation ratio of NPs at the small intestinal villi of mice. Data are presented as the mean ± SD ($n = 3$ independent experiments). *$P < 0.05$, **$P < 0.01$ and ***$P < 0.001$ by two-tailed Student's *t*-test. Source data are provided as a Source Data file. **e** Competitive adsorption of NPs in the intestinal villi of mice examined by CLSM; blue: nuclei of the intestinal villi stained with DAPI, green: FITC labeled NPs, red: RITC labeled NPs, scale bar: 100 μm. Experiment was repeated three times independently with similar results. **f** Relative fluorescent intensity of NPs in intestinal competitive adsorption. Source data are provided as a Source Data file. **g** Treatment protocols on the intestinal uptake study of NPs. **h** Bio-TEM images of the mucosa of mice at 4 h post oral administration of NPs. Experiment was repeated three times independently with similar results.

## Oral bioavailability and bio-distribution of NPs

In light of these positive results, the oral bioavailability of NPs was directly determined by measuring the Si concentration in the blood via inductively coupled plasma source mass spectrometer (ICP-MS). According to Fig. 5n, VSN and CVSN began to distribute in the blood at 4 h after administration, and still maintained high plasma concentrations even after 7 days, implying long blood circulation times. For SSN and MSN, the maximum plasma concentrations ($C_{max}$) were attained after 48 h of administration, and the plasma concentration showed a decreasing trend thereafter. By mimicking the shape, size, surface topology and chiral recognition process of viruses, CVSN achieved satisfactory oral absorption into the bloodstream with the largest area under the curve (AUC) and longer half-life ($t_{1/2}$, 253 h) than that of SSN and MSN (114.71 and 44.69 h, respectively).

NPs distribution in the major organs was also qualitatively and quantitatively analyzed by IVIS Lumina Series III Living Image System and ICP-MS, respectively. As shown in Fig. l−m and Supplementary Fig. 24, the fluorescence signals of CVSN were highest in almost all

organs, implying satisfactory bioavailability, while SSN and MSN were poorly distributed in organs. To be specific, in lung and brain, with abundant vascularity and numerous capillaries, the fluorescence intensity and Si concentration ranked as CVSN > VSN > MSN > SSN at 24 h and 7 day post administration (Fig. 5m, n). However, VSN were highly distributed in heart, spleen and liver at all time points, wherein the macrophage was rich, due to the passive targeting of NPs[45–47]. It should be noticed that, the liver was still the largest passive targeted organ for all the NPs owing to its large weight. Collectively, NPs exhibiting different surface topology showed distinguished distribution trends and biological fate, and CVSN exhibited the best oral adsorption and distribution characteristics.

## Cellular uptake of NPs

The internalization efficiency of NPs on Caco-2 cells was investigated via bio-TEM, CLSM and FCM (Supplementary Fig. 25). According to the direct observation via bio-TEM, spherical NPs (SSN and MSN) showed poor adhesion on the cell membranes with a single contact point,

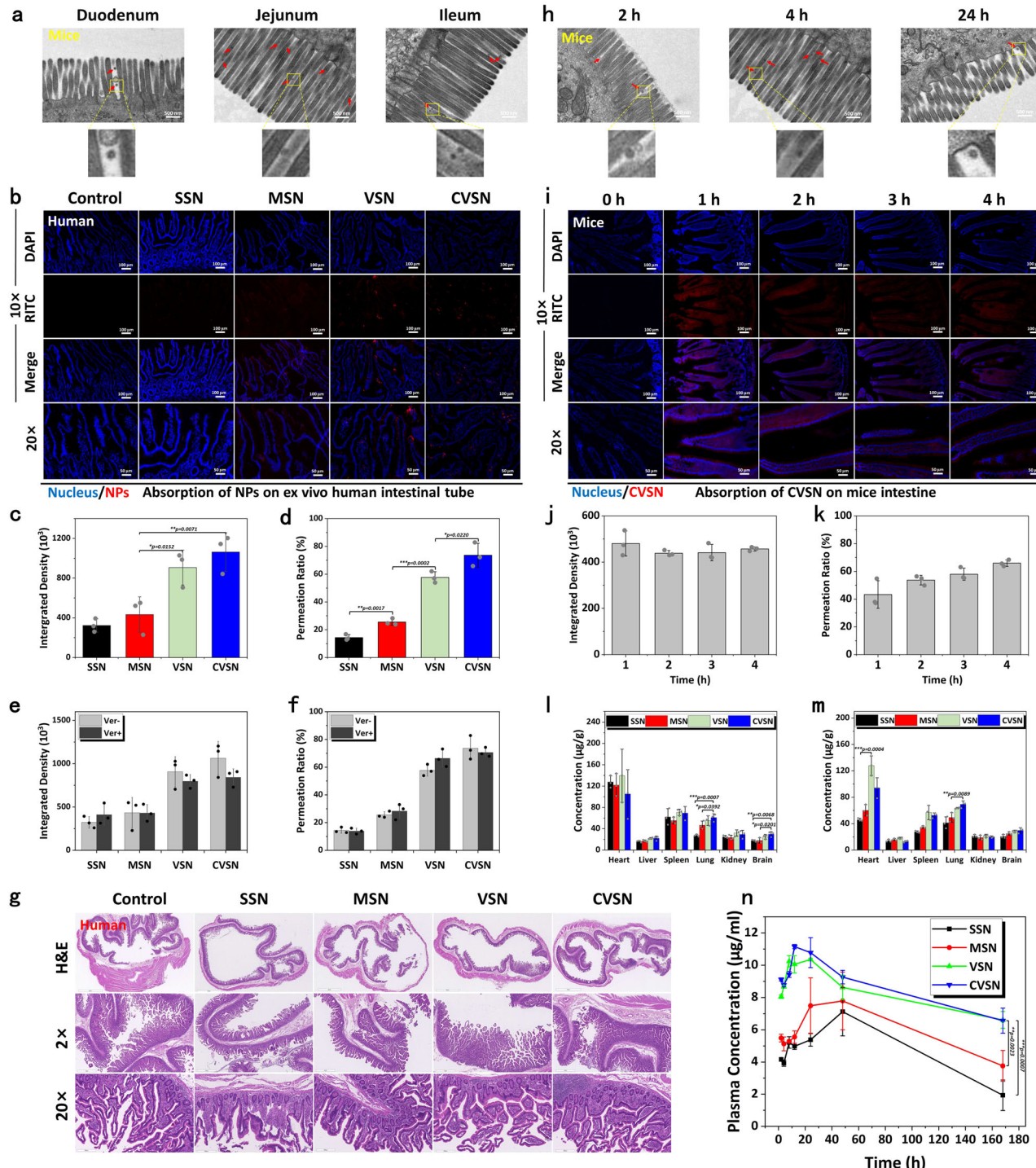

**Fig. 5 | Oral adsorption and distribution of NPs. a** Bio-TEM images of the mucosa from the duodenum, jejunum and ileum of mice at 4 h post oral administration of CVSN. Experiment was repeated three times independently with similar results. **b** CLSM images of NPs absorbed at the ex vivo small intestinal villi of human at 1.5 h post incubation without the per-treatment of verapamil (VER); blue: nuclei of the intestinal villi stained with DAPI, red: RITC labeled NPs. Experiment was repeated three times independently with similar results. **c** Fluorescence integrated density and **d** permeation ratio of NPs at the small intestinal villi of human. **e** Fluorescence integrated density and **f** permeation ratio of NPs at the small intestinal villi of human before and after pretreated with VER. All data of **c**–**f** are presented as the mean ± SD (*n* = 3 independent experiments). *P < 0.05, **P < 0.01 and ***P < 0.001 by two-tailed Student's *t*-test. Source data are provided as a Source Data file. **g** Hematoxylin and eosin (H&E) staining images of the small intestinal tissues of human after incubated with NPs. Experiment was repeated three times

independently with similar results. **h** Bio-TEM images of the mucosa from the jejunum of mice at 2, 4 and 24 h post oral administration of CVSN. Experiment was repeated three times independently with similar results. **i** Fluorescence microscope images of CVSN absorbed at the small intestinal villi of mice at 0, 1, 2, 3 and 4 h post oral administration; blue: nuclei of the intestinal villi stained with DAPI, red: RITC labeled CVSN. Experiment was repeated three times independently with similar results. **j** Fluorescence integrated density and **k** permeation ratio of CVSN at the small intestinal villi of mice at different times Plasma concentration of Si element in main organs at **l** 24 h and **m** 7 days after oral administration of NPs. **n** Time-dependent plasma concentration of Si element after oral administration of NPs. All data of **j**–**n** are presented as the mean ± SD (*n* = 3 independent experiments). *P < 0.05, **P < 0.01 and ***P < 0.001 by two-tailed Student's *t*-test. Source data are provided as a Source Data file.

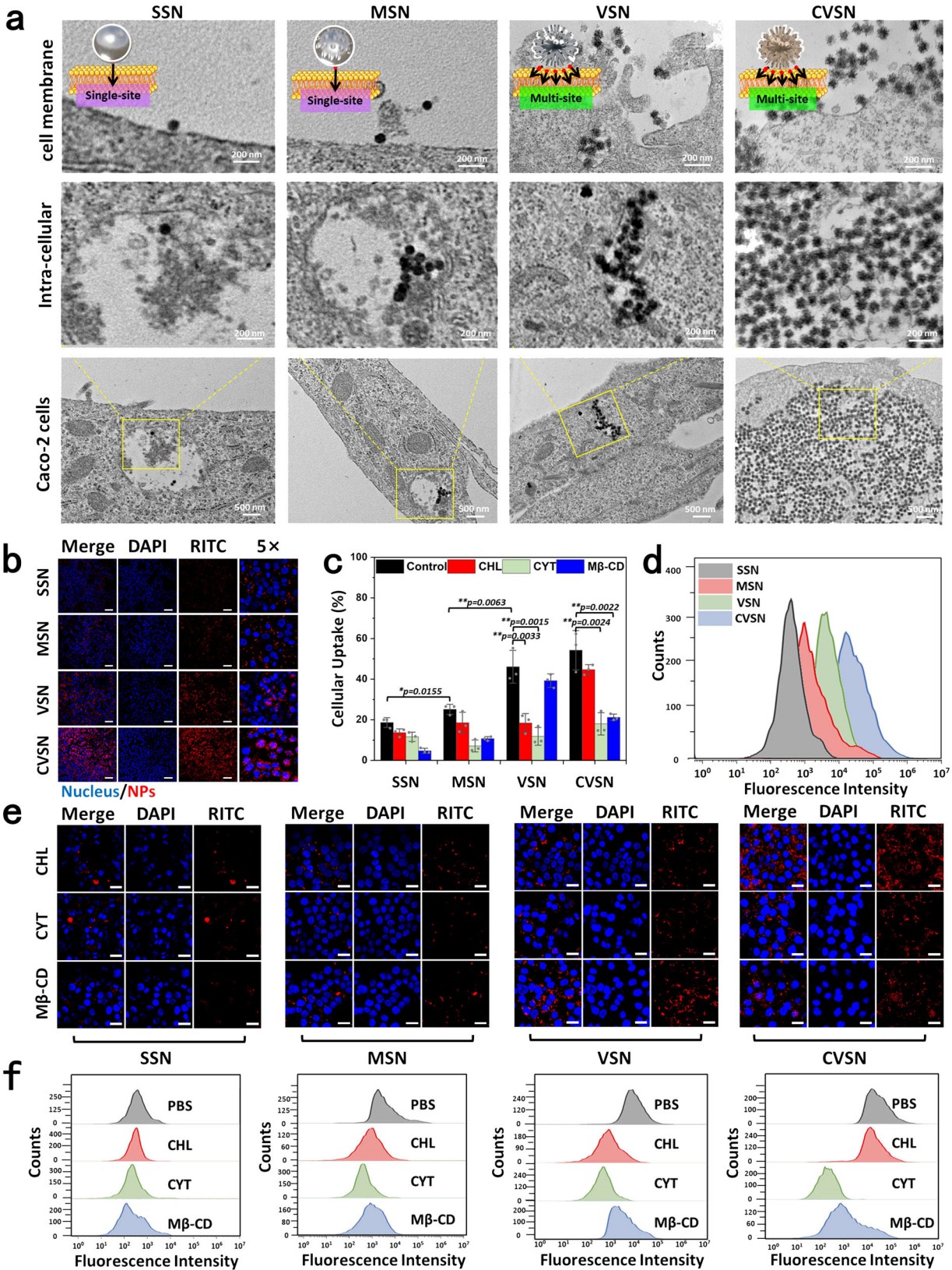

followed by the limited endocytosis into the cells (Fig. 6a). Conversely, VSN and CVSN with virus-mimic surface topology provided multiple contact sites and directly anchored on the cell membrane to facilitate cellular uptake. Excitingly, numerous CVSN appeared in the cells. CLSM images revealed that, SSN and MSN showed weak red fluorescent signals distributed far from nucleus (uptake amount 18.61% and

25.13%, respectively), whereas VSN and CVSN displayed strong fluorescent signals near the nucleus or intra nucleus, indicating efficient cellular internalization (Fig. 6b). The fluorescence intensity as well as the internalization efficiency ranked as CVSN > VSN > MSN > SSN (Fig. 6c, d and Supplementary Fig. 26). For virus-like NPs, the uptake started at 0.5 h post incubation and gradually moved toward the

**Fig. 6 | Cellular uptake of NPs on Caco-2 cells. a** bio-TEM images of the cellular uptake of NPs on Caco-2 cells. Experiment was repeated three times independently with similar results. **b** CLSM images of the cellular uptake of RITC labeled NPs on Caco-2 cells; blue: cell nucleus stained with DAPI, red: RITC labeled NPs, scale bar 100 μm. Experiment was repeated three times independently with similar results. **c** Cellular uptake amount (%) of NPs on Caco-2 cells before and after treatment with specific inhibitors semi-quantitative measured via the CLSM images. Data are presented as the mean ± SD ($n = 3$ independent experiments). *$P < 0.05$ and

**$P < 0.01$ by two-tailed Student's $t$-test. Source data are provided as a Source Data file. **d** Flow cytometry (FCM) analysis on the cellular internalization of NPs. **e** CLSM images of the cellular uptake of RITC labeled NPs on Caco-2 cells after pretreatment with specific inhibitors; blue: cell nucleus stained with DAPI, red: RITC labeled NPs, scale bar 20 μm. Experiment was repeated three times independently with similar results. **f** FCM analysis on the cellular internalization of NPs after pretreatment with specific inhibitors. Experiment was repeated three times independently with similar results.

nucleus after 4 h (Supplementary Fig. 27). We then compared the cellular internalization between CVSN and DVSN via FCM, and the results showed higher uptake amount for CVSN, confirming that the chirality of NPs still functioned at the cellular uptake phase (Supplementary Fig. 28). In a word, by simulating the viral infection (multi-sited anchor on the cell membranes and chiral recognized with the membrane proteins), significant number of CVSN rapidly entered the cells.

To elucidate the mechanisms for cellular internalization, cells were pretreated with chlorpromazine (CHL), cytochalasin D (CYT) and methyl-β-cyclodextrin (Mβ-CD), which were endocytosis markers for clathrin-mediated, macropinocytosis, and caveolae-mediated endocytosis, respectively. For SSN, pretreatment with CHL and CYT did not significantly restrict cellular uptake (inhibition rate: 26.68% and 37.57%, respectively), but reduced up to 74.60% uptake amount after co-incubation with Mβ-CD, which meant that SSN entered cells mainly through caveolae-mediated pathways (Fig. 6c, e). Meanwhile, CYT and Mβ-CD also observably decreased the internalization of MSN, basically suggesting the involvement of macropinocytosis endocytosis and caveolae-mediated endocytosis. In comparison, the pretreatment of CHL and CYT resulted in 60.07% and 74.31% inhibitions on the cellular uptake for VSN, implying the clathrinid-mediated and macropinocytosis pathways. By contrast, the use of Mβ-CD showed no significant (less than 15%) inhibitory effect. Different from that of VSN, the cellular uptake of CVSN mainly involved the macropinocytosis and caveolae-mediated pathways, and the pretreatment of CHL, CYT, and Mβ-CD inhibited the cellular uptake of 17.62%, 60.91%, and 66.81%, respectively. After pretreated with CHL, CYT and Mβ-CD, the inhibition rate measured to be 26.68%, 37.57%, and 74.60% for SSN, 26.36%, 71.72% and 57.48% for MSN, 56.50%, 79.31%, and 8.15% for VSN and 17.62%, 66.81% and 60.91% for CVSN, showing good agreement with the semi-quantitative data of CLSM (Fig. 6f and Supplementary Fig. 26).

## Biocompatibility of NPs

The biocompatibility of NPs was estimated both in vitro and in vivo. First, the cytotoxicity of NPs was evaluated by CCK-8 assay. As shown in Supplementary Fig. 29, there was no significant effect on the vitality of cells after 24, 48, and 72 h incubation of NPs, and the survival rates were all above 85% and not obviously dependent on the amount of NPs, demonstrating good cytocompatibility. As one of the key issues concerned with the safety of biomaterials, the hemocompatibility of NPs was evaluated by hemolysis and protein adsorption assays. The results showed that, the hemolysis rates (<6%) of NPs ranging from 5–100 μg/ml were all concentration-dependent but in accordance with the standard for biomaterials: ISO 10993 – 4:2002.40 (Supplementary Fig. 30). In addition, the hemolysis rate slightly decreased after the chiral modification of L-Ala. Moreover, the bovine serum albumin (BSA) adsorption of NPs was all <10%, indicating good hemocompatibility (Supplementary Fig. 30). Compared to VSN, the increased protein adsorption by CVSN might be due to the chiral recognition.

To detect the underlying in vivo toxicity of NPs, mice exposed to SSN, MSN, VSN, or CVSN were executed on the 14th day, and their blood and major organs were collected and subjected to hematological and biochemical tests, and histopathological examinations,

respectively (Supplementary Fig. 30). No sudden death, unusual behaviors, and significant weight loss were observed during the experimental period. The organ/body ratios (%) of the major tissues (heart, liver, spleen, lung, and kidney) appeared to be normal. H&E staining images revealed that SSN, MSN, VSN, and CVSN would not cause obvious histopathological abnormalities or damage after oral administration. Furthermore, the hematological and biochemical parameters were all within the reference ranges and showed no substantial difference between the groups. Collectively, the NPs lacked any indication of toxicity both in vitro and in vivo.

## Drug loading and release properties of NPs

Inspired by the core–shell structure of viruses, which could package and protect the payload, MSN, VSN, and CVSN with mesoporous structures were selected for the delivery of IMC. Based on the solvent evaporation method, the drug loading amount of IMC@MSN, IMC@VSN and IMC@CVSN were 21.23%, 19.55%, and 21.30 %, respectively (Supplementary Fig. 31). As illustrated in Fig. 7a, the FTIR spectra of IMC presented the characteristic absorption bands at 1717.5 and 1691.2 cm$^{-1}$, referred to the -C = O group and the -NH group, respectively. Conversely, IMC@MSN, IMC@VSN, and IMC@CVSN showed similar spectra to those of the silica NPs with the disappearance of most IMC characteristic bands, confirming the incorporation of IMC. N$_2$ adsorption/desorption analysis also demonstrated the successfully loading of IMC, as evidenced by the degeneration of isotherms and the decrease in texture parameters of NPs (including S$_{BET}$ and pore diameter), owing to the occupation of nanopores (Supplementary Fig. 31 and Supplementary Table 1). Furthermore, the XRD and differential scanning calorimetry (DSC) studies were conducted to determine the transformation of the crystalline state before and after drug loading. MSN, VSN, and CVSN showed broad XRD diffraction bands between 5° to 40°, indicating their amorphous nature, whereas IMC was highly crystalline and had sharp diffraction peaks (Fig. 7b). After drug loading, IMC underwent an amorphization process along with the variation of peak pattern from sharp diffraction peaks to broad bands, which were quite different from those of the physical mixtures. Moreover, the DSC curves of IMC@MSN, IMC@VSN, and IMC@CVSN exhibited the absence of endothermic event, illustrating that IMC existed in an amorphous state owing to the limited space (Supplementary Fig. 31). In addition, the solid state stability of the Nano-DDS was confirmed by the XRD results after storing at ambient temperature and pressure for 30 days.

Prior to the drug release study, the wettability of Nano-DDS was evaluated to provide insights from the surface/interface properties. Compared to hydrophobic IMC, the wettability of IMC@MSN, IMC@VSN, and IMC@CVSN were greatly improved with reduction in the initial contact from 66.09° to 35.26°, 27.18°, and 23.75°, respectively (Supplementary Fig. 31). Notably, CVSN contributed to the highest improvement on the wettability of insoluble drugs. Next, the drug release of IMC@MSN, IMC@VSN, and IMC@CVSN was analyzed by employing pure IMC as a reference. The results showed that the release rate and release amount of IMC significantly increased with the assistance of NPs as a consequence of the change on drug crystalline state (from the crystalline to amorphous state; Fig. 7c, d and Supplementary Fig. 31). IMC@MSN, IMC@VSN, and IMC@CVSN can enhance

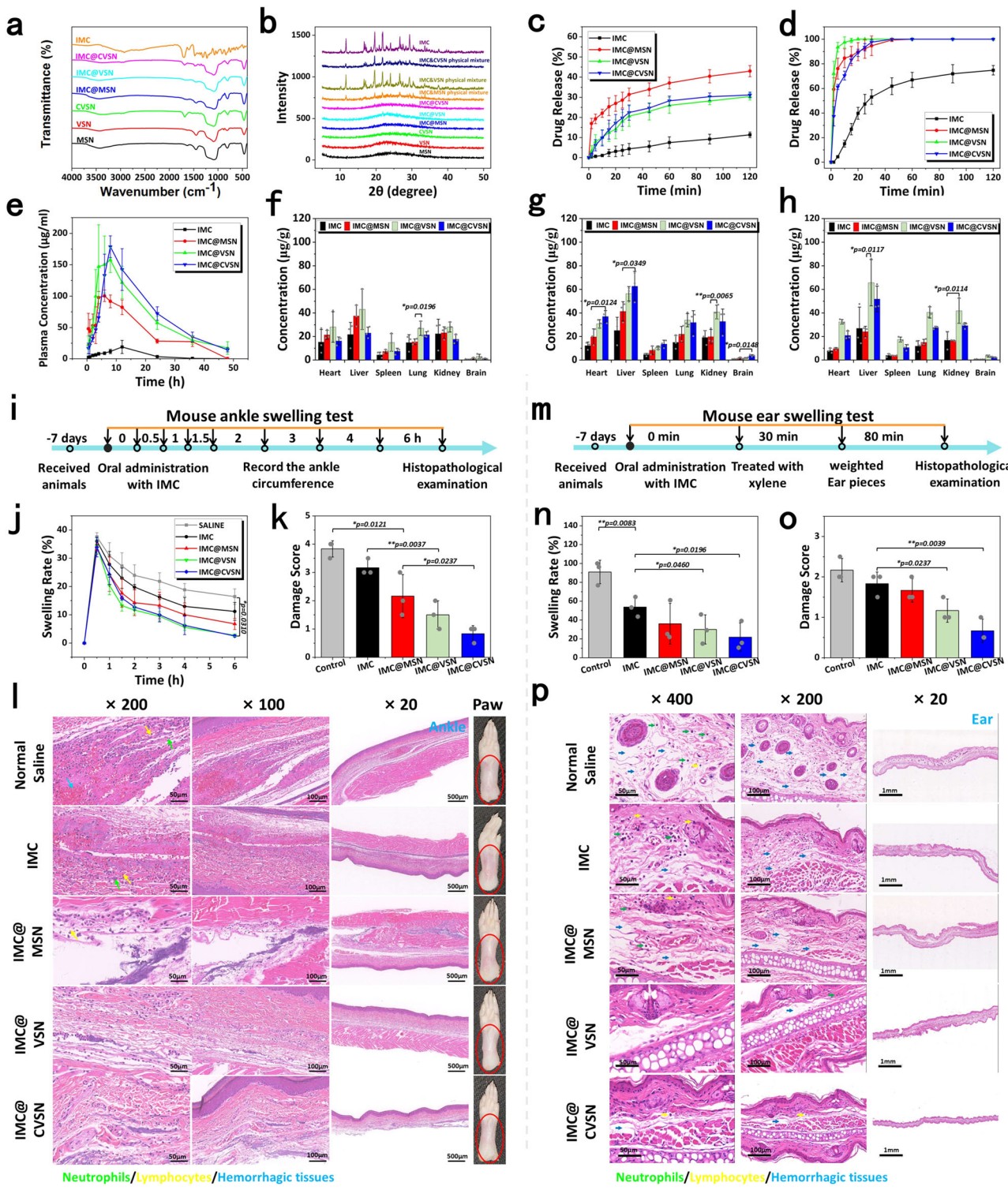

Neutrophils/Lymphocytes/Hemorrhagic tissues

the release of IMC from 11.35% to 42.99%, 30.38%, and 31.14% in SGF over 2 h, respectively. By the way, the drug release of both IMC and the Nano-DDS increased with pH. Particularly, the release rates of IMC@VSN and IMC@CVSN reached 100% at 90 min in SIF, facilitating the passive absorption of the payload.

## Oral adsorption and bio-distribution of the IMC loaded NPs

We then assessed the oral delivery of the IMC loaded NPs by systematically analyzing the absorption, distribution, and metabolism of drug. First, the oral bioavailability of IMC loaded NPs were directly evaluated by the everted intestinal sacs ex vivo and the

pharmacokinetics study in vivo. Supplementary Fig. 32 showed that the intestinal transport for IMC@CVSN was highly efficient, wherein the transport amount of IMC@CVSN was up to 7.70-times higher than that of IMC. On the other hand, IMC@MSN, IMC@VSN and IMC@CVSN shorten the peak concentration time ($T_{max}$) of IMC from 12 h to 6, 4 and 8 h, respectively, and the maximum blood concentrations ($C_{max}$) followed the order of IMC@VSN ≈ IMC@CVSN > IMC@MSN > IMC (Fig. 7e and Supplementary Table 2). In accordance with the oral adsorption efficiency of NPs, IMC@CVSN showed satisfactory oral bioavailability (1145.91%) by simulating the shape, size, surface topology and gene package of viruses. Drug concentrations in

**Fig. 7 | Oral delivery efficiency of IMC loaded NPs. a** FTIR spectra and **b** X-ray diffraction (XRD) patterns of IMC@MSN, IMC@VSN, and IMC@CVSN before and after drug loading. Source data are provided as a Source Data file. In vitro drug release of IMC loaded NPs in **c** SGF and **d** SIF. Data are presented as the mean ± SD ($n = 3$ independent experiments). Source data are provided as a Source Data file. **e** Plasma concentration-time profiles of IMC loaded NPs. Data are presented as the mean ± SD ($n = 3$ independent experiments). Source data are provided as a Source Data file. Distribution of IMC loaded NPs in major organs at **f** 1 h, **g** 3 h and **h** 6 h after oral administration. Data are presented as the mean ± SD ($n = 3$ independent experiments). *$P < 0.05$ and **$P < 0.01$ by two-tailed Student's $t$-test. Source data are provided as a Source Data file. **i** Treatment protocols on the mouse ankle swelling test (MAST). **j** The anti-inflammatory effects of IMC loaded NPs on MAST. Data are presented as the mean ± SD ($n = 3$ independent experiments). *$P < 0.05$ by two-tailed Student's $t$-test. Source data are provided as a Source Data file. **k** Damage score on the histopathological images of MAST. Data are presented as the mean ± SD ($n = 3$ independent experiments). *$P < 0.05$ and **$P < 0.01$ by two-tailed Student's $t$-test. Source data are provided as a Source Data file. **l** Representative appearances photographs and histopathological images on the right hind paw of rats, (the green, yellow and blue arrows marked neutrophils, lymphocytes and hemorrhagic tissues, respectively). Experiment was repeated three times independently with similar results. **m** Treatment protocols on the mouse ear swelling test (MEST). **n** The anti-inflammatory effects of IMC loaded NPs on MEST. Data are presented as the mean ± SD ($n = 3$ independent experiments). *$P < 0.05$ and **$P < 0.01$ by two-tailed Student's $t$-test. Source data are provided as a Source Data file. **o** Damage score on the histopathological images of MEST. Data are presented as the mean ± SD ($n = 3$ independent experiments). *$P < 0.05$ and **$P < 0.01$ by two-tailed Student's $t$-test. Source data are provided as a Source Data file. **p** Histopathological images on the ear of mice, (the green, yellow and blue arrows marked neutrophils, lymphocytes and hemorrhagic tissues, respectively). Experiment was repeated three times independently with similar results.

the main organs after oral administration were shown in Fig. 7f–h. Generally speaking, IMC presented poor oral adsorption and bio-distribution, and was rapidly metabolized by the liver and kidney. However, these features were substantially improved after loading into the NPs. Especially for IMC@CVSN, IMC displayed significantly increased accumulations in all main organs, further confirming the satisfactory bioavailability. IMC@VSN was rapid distributed in the main organs at 1 h post administration, whereas IMC@CVSN had the highest in vivo accumulation at 3 h. These results provided preliminary evidences for the efficiently topology-dependent delivery advantages of CVSN to ameliorate the drug adsorption in vivo.

### Anti-inflammation pharmacodynamics of IMC loaded NPs

MAST, MEST and mouse writhing test (MWT) were carried out to evaluate the anti-inflammatory effects of IMC loaded NPs. Direct observations revealed that, severe swelling occurred in the rats' ankles in the saline group, while slight swelling relief was observed in the IMC treated groups due to the anti-inflammatory effect of the drug (Fig. 7i–k and Supplementary Table 3). After 6 h, the swelling rates were 16.41%, 11.18%, 6.76%, 2.77%, and 2.49% for animals in the saline, IMC, IMC@MSN, IMC@VSN and IMC@CVSN groups, respectively. The subcutaneous tissues in the saline group were hemorrhagic and edematous with the infiltration of inflammatory cells (including neutrophil infiltrates [green arrows] and lymphocytic infiltrates [yellow arrows], and the disease damage score was 3.83; Fig. 7k, l). Histopathological examinations on the animals treated with IMC and IMC@MSN also discovered hemorrhage, edema and inflammatory cell infiltration in the subcutaneous and muscular tissues. Conversely, only a small number of inflammatory cells and slight infiltration were observed in the IMC@CVSN groups. In this case, the damage score reduced to 0.83. Conformance to the swelling rates, the damage scores followed the order of negative control (animals treated with normal saline) > positive control (animals treated with IMC) > IMC@MSN > IMC@VSN > IMC@CVSN, further verifying the improvement of bioavailability. By the way, the NPs also helped to reduce the irritation of NSAID on the gastric and intestinal mucosa by avoiding the direct contact with IMC crystals (Supplementary Fig. 33)

The results of the MEST were similar to that of MAST. In details, the ear swelling rate of the control group was 90.96%, which confirmed the successful induction of swelling by xylene (Fig. 7m, n). The swelling rates in IMC (positive control), IMC@MSN, IMC@VSN, and IMC@CVSN groups were 53.77%, 36.06%, 29.98%, and 21.79%, respectively. The same trends were also evident in the pathological slides of the ear tissues. The ear thickness increased significantly in animals in the control group, and large areas of edematous (blue arrow) and eosinophils (green arrow), along with a small number of lymphocytes (yellow arrow), were observed in the pathological slides, and the damage score rated as 2.17. After the pre-treatment of IMC, severe edematous and abundant eosinophils were still evident. Upon

oral administration of IMC@MSN, IMC@VSN, and IMC@CVSN, the microscopic disease damage score decreased in turn as a consequence of the improved bioavailability. Particularly, a non-significant increase in ear thickness was observed in animals in IMC@CVSN groups, with slight edema and eosinophilic infiltration (damage score was 0.67; Fig. 7o, p).

In addition, to assess the analgesic effects of IMC loaded NPs, pain model was inducted using acetic acid. In the negative control group, mice twisted 27.00 times in 20 min. IMC, IMC@MSN, IM@VSN and IMC@CVSN could reduce the writhing number to 18.33, 10.33, 6.67, and 7.00, respectively (Supplementary Fig. 34). The inhibition rates of IMC@VSN and IMC@CVSN reached 234.62% and 230.77%, respectively (Supplementary Table 4).

### Versatility of CVSN as oral Nano-DDS

Finally, to verify the versatility of CVSN as oral Nano-DDS, a series of NSAIDs, including nimesulide (NMS), acetaminophen (AC), aspirin (ASP), celecoxib (CEL), flurbiprofen (FB), ibuprofen (IBU) and IMC belonging to BCS 2–4 (characterized by poor solubility and/or low permeability) were encapsulated into CVSN (Fig. 8a, b). Excitingly, these NSAIDs with diverse structures, molecular weights and function groups were all efficient loaded into the nanopores of CVSN (drug loading amount: 20.53–29.53%), along with the amorphization of drug (Fig. 8d, Supplementary Fig. 35 and Table S5). Subsequently, we rapidly investigated the ex vivo intestinal transport and the in vivo bioavailability of NMS@CVSN, AC@CVSN, ASP@CVSN, CEL@CVSN, FB@CVSN, IBU@CVSN, and IMC@CVSN via everted intestinal sacs models and MEST models, respectively. As except, CVSN was able to improve the intestinal transport of almost all NSAIDs, wherein the transport rate of NMS-, AC-, ASP-, CEL-, FB-, IBU- and IMC-loaded CVSN were up to 1.63, 2.14, 2.40, 3.12, 3.04, 1.60, and 7.70-times higher than that of the naked drug at 4 h (Fig. 8c, e–i). These Nano-DDS also achieved satisfactory therapeutic effects, and the inhibition rates were 141.00%, 208.28%, 121.12%, 125.45%, 148.88%, 142.89%, and 229.95% for NMS@CVSN, AC@CVSN, ASP@CVSN, CEL@CVSN, FB@CVSN, IBU@CVSN and IMC@CVSN, respectively (Fig. 8m, n and Table S6). Moreover, H&E staining of swelling tissues also confirmed the superiority of CVSN in oral delivery, in which large areas of edematous, hemorrhages (blue arrow) and granulocytes (green arrow) were noticed in the control and NSAIDs groups (the damage score was 3.83–3.33), whereas the ear swelling was largely relieved after pre-administration with these Nano-DDS. Only slight edema (blue arrow) and granulocytes (green arrow) were found in the Nano-DDS groups, and the disease damage score decreased to 2.33–0.83 (Fig. 8o, p).

### Discussion

Considering the extremely high infectivity and penetrability of viruses, they have become valued inspiration sources for the design of novel Nano-DDS in recent years. Various virus-mimicking strategies with the

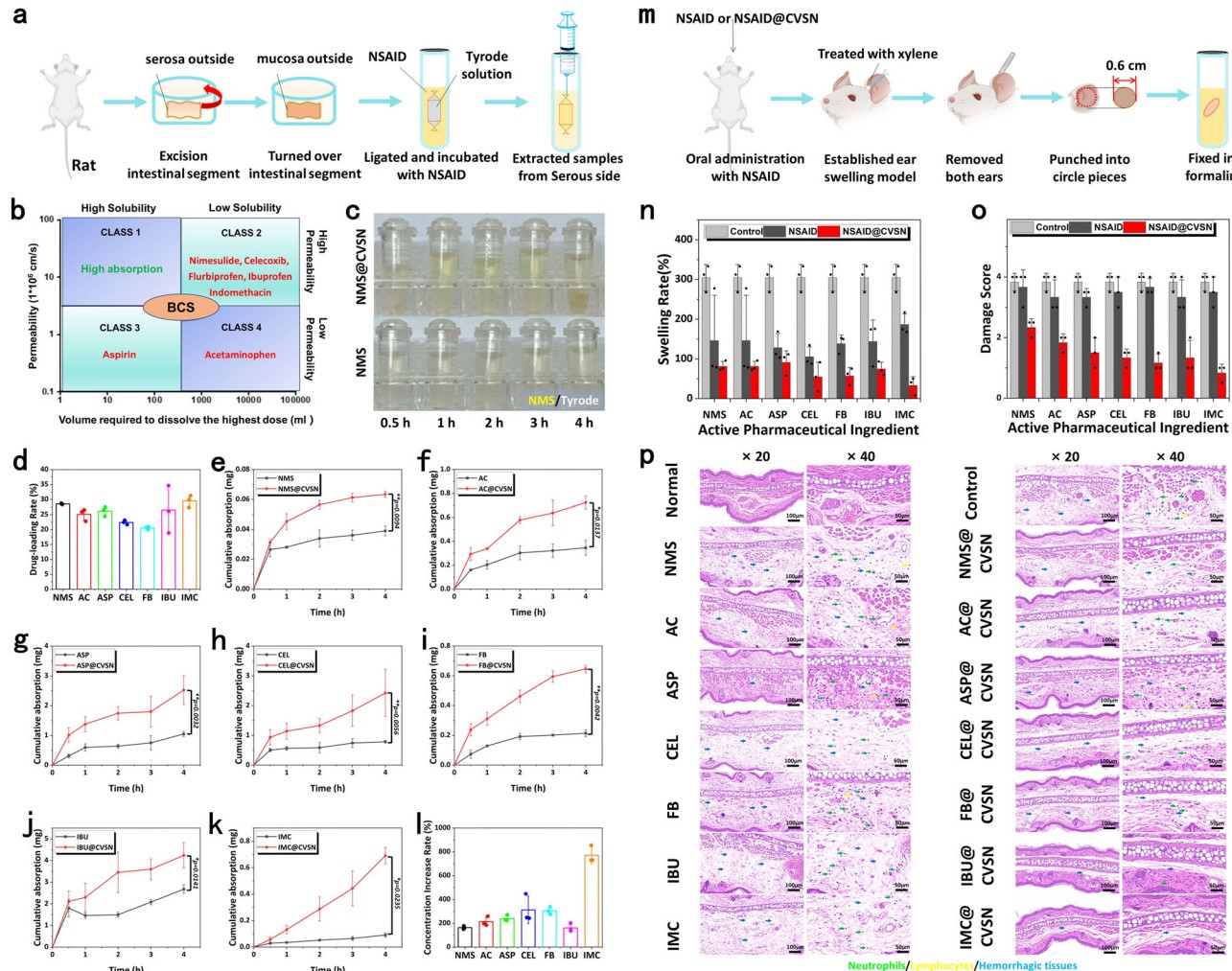

**Fig. 8 | Versatility of CVSN as oral Nano-DDS. a** Treatment protocols on the everted intestinal sacs model. **b** BCS classification of the NSAIDs. **c** Images of NMS and NMS@CVSN samples collected from the serosa side of everted intestine sac. Experiment was repeated three times independently with similar results. **d** NSAIDs loading capacity for CVSN. Data are presented as the mean ± SD (*n* = 3 independent experiments). Source data are provided as a Source Data file. Transport amount of **e** NMS, **f** AC, **g** ASP, **h** CEL, **i** FB, **j** IBU and **k** IMC measured by everted intestinal sacs model. Data are presented as the mean ± SD (*n* = 3 independent experiments). *$P < 0.05$ and **$P < 0.01$ by two-tailed Student's *t*-test. Source data are provided as a Source Data file. **l** Increase rate on the cumulative intestinal transport amount of

drug after loading into CVSN. Data are presented as the mean ± SD (*n* = 3 independent experiments). Source data are provided as a Source Data file. **m** Treatment protocols on the MEST. **n** The anti-inflammatory effects of NSAIDs loaded NPs on MEST. Data are presented as the mean ± SD (*n* = 3 independent experiments). Source data are provided as a Source Data file. **o** Damage score on the histopathological images of MEST. Data are presented as the mean ± SD (*n* = 3 independent experiments). Source data are provided as a Source Data file. **p** Histopathological images on the ear of mice, (the green, yellow and blue arrows marked granulocytes, lymphocytes and hemorrhagic tissues, respectively). Experiment was repeated three times independently with similar results.

purpose of efficient drug/gene delivery have been developed, including the direct utilizations of VNPs and VLPs, the simple imitations on the sizes, shapes and core–shell structures of viruses, the application on the functional proteins of viruses, and the deep simulations on the transfection modes, stepwise responses and immune activations of viruses.

In this study, we developed an efficient oral Nano-DDS CVSN by mimicking both the structure and function of viruses, wherein core–shell mesoporous silica NPs (~80 nm) with virus-like nanospikes was fabricated to simulate the morphology of viruses, and molecular chiral moiety was modified to enable chiral recognition. By comparing with the NPs exhibiting essentially the same structural parameters but different surface topology, including SSN, MSN and VSN, the delivery advantages of CVSN in overcoming intestinal mucosal barrier were listed as follows: (1) viral surface topology as well as molecular chiral modification increased the surface roughness and interface activity of NPs; (2) the external nanospikes provided abundant binding sites to

the bio-membrane to improve the bio-adhesion and transmembrane capacity; (3) CVSN with molecular chirality could chiral recognize with the membrane protein and facilitate the endocytosis. As a result, CVSN showed strongest bio-adhesion ability, longest GI tract retention time, best mucus permeability, highest cellular uptake, efficient oral adsorption, and favorable bio-distribution via multi-sited anchoring and active chiral recognition. From the perspective of functional bionics, drugs were loaded into the NPs to simulate the package of genes, wherein the payloads were bounded into the nanopores in the amorphous state and isolated from the biological environment to improve their stability and solubility, while the chiral nanospikes multi-sited anchored and chiral recognized on the intestinal mucosa to enhance the penetrability. It was expected that, by mimicking the shape, size, surface topology, chiral recognition and gene encapsulation of viruses, CVSN was proven to be a versatile Nano-DDS for BCS 2-4 drugs to efficiently overcome the multiple physiological barriers in the GI tract and achieve favorable oral drug delivery efficiency. In one word, both

the in vitro and in vivo experiments verified the multiple functionality, high stability, low irritation, negligible toxicity, and good biodegradability and biocompatibility of CVSN, pathing important research value and clinical translation prospects. We believe that the virus-mimic strategy as well as the explorations on the structure–function relationships of NPs would provide valued inspiration sources for the design of novel oral Nano-DDS.

## Methods

### Animals, patient samples and cells

Our research complies with all relevant ethical regulations. All animal experiments were approved by the Animal Ethics Committee of China Medical University (Protocol Number: CMUKT2022255). Institute of Cancer Research (ICR) mice (male, 6 weeks old, ~30 g), and Sprague-Dawley (SD) rats (male, 6 weeks old, ~200 g) were involved in the study, and received humane care in accordance with the Guide for the Care and Use of Laboratory Animals. Animals were housed in equipped animal facility at a constant temperature (~25 °C) under a dark/light cycle, treated with a standard diet and water, and fed adaptively for 1 week after arrival. Although we have used single-sex animals in our research, we think that the research results were not only applicable to single sex. The collection of ex vivo patient samples was approved by the Ethics Committee of the China Medical University (Protocol Number: [2022]436, [2023]64). Paraneoplastic tissues were obtained from 20 Chinese small intestinal stromal tumor (SIST) patients (male or female [including 12 male and 8 female, we determined the sex and/or gender of participants based on the record in medical records], aged larger than 18, who informed consented to donate their biological samples) at the First Hospital of China Medical University. This study did not involve clinical trials or disease research, sex and gender were identified as insignificant factor in results analysis. Caco-2 cells were provided by the Procell Life Science Technology Co. Ltd (CL-0005) and cultured in Dulbecco's modified Eagle medium (DMEM) containing 10% fetal bovine serum (Hy Clone, USA). Cell lines were validated using short tandem repeat (STR) markers and were tested negative for mycoplasma contamination.

### Software and code

The software used for data collection in this study includes Zetasizer Software7.13; APPCIMain 8.60.160.11228; Brucker MI SE; Aperio Image Scope v12.3.3.3.5048; Slide Viewer; ZEN 3.6(ZEN lite) v3.6.095.06000 and BD Accuri C6 Plus Software1.0.23.1. The software used for data analysis in this study includes Origin 2021 9.8.0.200; Image J1.53e; DAS v2.0; Microsoft Office 2016; FlowJo_v10.8.1. No codes were involved in this study.

### Experimental reagents

Cetyltrimethylammonium bromide (CTAB), 3-aminopropyltriethoxysilane (APTES), tetraethoxysilane (TEOS) and L/D-Boc-Ala were purchased from Shanghai Aladdin Biotechnology Co., Ltd. 1-(3-Dimethylaminopropyl)-3-ethylcarbodiimide hydrochl (EDCI), 1-hydroxybenzotrizole (HOBT) and trifluoroacetic acid (TFA) were purchased from Shanghai Rhawn Chemical Technology Ltd. RITC, FITC and DAPI were provided by Beijing Bailingway Technology Co., Ltd. IMC was purchased from Shanghai Maclean Biochemicals Co., Ltd. University of Wisconsin (UW) solution was obtained from Preservation Solutions. Tyrode's solution was provided from Shanghai Yuanye Biotechnology Co., Ltd. Double deionized water was produced by ion exchange. All other chemicals were of reagent grade and were used without further purification.

### Synthesis of NPs

SSN, MSN and VSN were synthesized according to a previously reported method with some modifications[37]. For the synthesis of SSN,

2 ml TEOS was added dropwise to a mixed solution containing 40 ml ethanol, 3 ml deionized water and 0.8 ml ammonia, and stirred for 12 h. The product was collected by centrifugation, washed several times with water and ethanol, and thoroughly dried.

For the synthesis of MSN, 1 g CTAB and 1 ml ammonia were introduced into the mixed solution of 100 ml deionized water and 30 ml ethanol, and stirred until dissolution. Then 3 ml TEOS was dropwise added to the system, stirred for 4 h, and stood statically for 24 h. The product was collected by centrifugation, washed with water and ethanol, and dried. After that, the final product named MSN was obtained by calcining in a muffle furnace at 550 °C for 6 h to remove the template CTAB.

To prepare VSN, 1 g CTAB and 0.18 g TEA were added into 40 ml deionized water and stirred at 60 °C for 2 h. Then 20 ml mixed solution of cyclohexane and TEOS (4:1, v/v) was dropwise added, stirred at 60 °C for 48 h, and transferred to an oil bath at 98 °C for 24 h. After that, the product was collected by centrifugation, washed with water and ethanol, and dried. The final product VSN was obtained by calcining in a muffle furnace at 550 °C for 6 h.

To prepare CVSN, we first accomplished the amination of VSN (denoted as AVSN) by using APTES. In a typical run, 100 mg VSN and 300 μl APTES were dispersed in 30 ml anhydrous ethanol and stirred at 50 °C for 24 h. The sample was washed with ethanol, centrifuged, dried and collected to obtain AVSN. Then, the chiral group L-Ala was modified onto AVSN via an acylation reaction. 20 mg AVSN was dispersed in 5 ml anhydrous DMSO. Then 82.34 mg EDCI, 58.03 mg HOBT and 81.27 mg Boc-L-Ala were added to the above mixture and stirred for 48 h. Afterward, the product was centrifuged, washed alternately with water and ethanol, and dried to obtain Boc-L-AVSN. To remove the Boc protecting group, 15 mg Boc-L-AVSN was ultrasonically suspended into 15 ml DMSO followed by the addition of 6.53 ml TFA, and the reaction was carried out at room temperature for 3 h under stirring. Afterwards, the system was centrifuged, washed with ethanol and thoroughly dried to obtain CVSN. For comparison, we also synthesized DVSN through the same procedure except that Boc-D-Ala was used instead of Boc-L-Ala.

### Characterization of NPs

FTIR spectra of NPs were recorded by an infrared spectrometer (Spectrum 1000, Perkin Elmer, USA) at 400–4000 cm$^{-1}$ using the pressed KBr tablet method. The DLS size distribution of NPs was investigated with a zeta sizer (Malvern Zeta-sizer NanoZS90). The morphologies of the NPs were studied by TEM (FEI Tecnai G2-F30, USA), SEM (JEOL JSM-6510 A, Japan) and AFM (Oxford cypher ES, USA), respectively. The nitrogen adsorption/desorption tests were performed using nitrogen adsorption analyzer (V-Sorb 2800 P, Gold APP, China) to obtain the pore texture parameters. The $S_{BET}$ and average pore size ($V_t$) of the NPs were calculated by Brunauer−Emmett−Teller (BET) and Barrett−Joyner−Halenda (BJH) methods, respectively. To study the mesoscopic structure, NPs were subjected to the SAXS analysis on an X-ray scattering instrument (D8 ADVANCE, BRUKER, Germany) at the angles (2θ) of 0.1–5°. Moreover, TGA curves of SSN, MSN, VSN, and CVSN were recorded with a microcomputer differential thermal analysis instrument (Q1000, TA Instrument, USA) in the range of 20–700 °C. CD spectra of CVSN were analyzed by comparing with VSN and DVSN, which was synthesized by grafting D-Ala onto the surface of VSN, in the range of 190–250 nm using a multifunctional CD spectrometer (MOS-500 from Bio-Logic, France).

### Surface/interface properties

To evaluate the surface/interface properties of the NPs, a series of studies, including the $D_{SSH}$, the wettability, the $P_{ow}$, the amino acid adsorption capacity, and the degradation property studies were carried out.

First, $D_{SSH}$ of NPs was calculated by measuring the weight loss acquired by TGA and the $S_{BET}$ according to following formula.

$$D_{SSH} = \frac{W_H}{S_{BET}} \times \frac{N_A}{18} \qquad (1)$$

Where $W_H$, $S_{BET}$ and $N_A$ denote the weight loss of NPs (from 200–700 °C), the $S_{BET}$ of samples and the Avogadro's number, respectively.

The wettability of NPs was evaluated by measuring the initial and equilibrium contact angles. In a typical run, SSN, MSN, VSN, and CVSN (60 mg, $n = 3$) was respectively pressed into a tablet, placed on a contact angle tester (model JCY series, Fangrui, China), and exposed to a drop of water. At the initial moment of the contact, the images were taken continuously at 5 s/frame, and the contact angle was measured using the tangent method.

The $P_{ow}$ of NPs was studied via the shaking method to evaluate their hydrophilicity. Before analysis, 100 ml octanol and 300 ml water were shaken at 30 °C for 24 h to prepare the saturated solutions. RITC labeled NPs was dispersed in 1 ml octanol-saturated water and placed in a syringe. Then 4 ml water-saturated octanol was added, vortexed for 3 min and shaken at 30 °C for 1 h. The NPs distributed in the aqueous and organic phases were evaluated by measuring the fluorescence intensity via a microplate reader (Bio-Rad Laboratories Ltd., Hertfordshire, UK) and calculating the $P_{ow}$.

To evaluate the chiral recognition potential, the amino acid adsorption capacity of CVSN was studied by comparing with DVSN. 15 mg CVSN ($n = 3$) was respectively added into 1 ml L-Ala or D-Ala solution (400 µg/ml), sealed and shaken at 37 °C for 12 h. After centrifugation, the supernatant was taken for CD testing. Similarly, the CD spectra after the incubation of DVSN with L-Ala or D-Ala solution were also recorded.

## Mucus permeation

Small intestines were collected from Sprague-Dawley (SD) rats (weighing 200 ± 20 g) following fasting overnight, and gently washed with SIF to remove the contents. A clean intestinal segment with 2 cm in length was intercepted, ligated at both sides, stained with 500 µl FITC solution (10 µg/ml, inject into the intestinal lumen using a syringe), and incubated in Tyrode's solution at 37 °C for 15 min. Afterward, the residual FITC solution was discarded, and the intestinal segment was injected with RITC labeled NPs (200 µl, 100 µg/ml) and incubated in Tyrode's solution at 37 °C for 30 min. The intestine samples were then placed on a culture dish (keeping the lumen side downward) and imaged by using a 63× oil microscope in CLSM (TI-E + C2 SI, Nikon C2Si, China) at z-stack mode, with images taken at a thickness of 50 µm and reconstructed in 3D, and quantified the fluorescence intensity by Image J software.

Furthermore, fresh intestinal mucus was carefully collected from intestinal segments of rats and human. To simulate particle penetration in mucus, 1.5 ml mucus was added to a 5 ml syringe, and then 0.5 ml SIF suspension of RITC labeled NPs (0.5 mg/ml, $n = 3$) were slowly injected into the top layer of mucus and allowed to freely diffuse into the mucus. After incubation at 37 °C for 24 h, photographs were taken to record the permeation capacity; and the NPs at the lower layer (0.5 ml) of syringe was removed, washed, dried, suspended in SIF, and measured the fluorescence intensity to calculate the permeation ratio.

## Bio-adhesion on mucosal tissues

First, fresh small intestine was collected from rats, gently washed, and cut into pieces of 1 × 3 cm. The bio-adhesion of NPs on the mucosal tissues was evaluated via the elution device. In a typical run, the mucosal tissues were smeared with NPs (20 mg, $n = 3$), incubated in a sterile incubator at 37 °C for 20 min. and placed on a platform fixed at

45°. Subsequently, NPs were eluted from the small intestinal tissues by SIF at the flow rate of 2 ml/min, collected every 1 min, centrifuged, gently washed to detach the adsorbed ions, thoroughly dried, and weighed to measure the bio-adhesion ratio, which was calculated using the following equation:

$$Bio - adhesion\,Ratio\,(\%) = \frac{W_0 - \sum_{i=1}^{5} W_i}{W_0} \times 100\% \qquad (2)$$

Where $W_O$ and $W_i$ denote the initial weight of samples and the weight of collected sample at time t, respectively.

Next, 20 mg RITC labeled NPs was spread evenly on the mucosal tissues and eluted by SIF. Then the mucosal tissues were subjected to fluorescent imaging using IVIS Lumina Series III Living Image System (Caliper Life Sciences, USA), and captured the region of interest (ROI) values from the flux in total photon count per cm².

Finally, taking into consideration the heterogeneity of species, bio-adhesion study was also carried out on ex vivo human small intestinal tissues. Fluorescent NPs (single NPs or pre-mixed NPs labeled with different fluorescein) was smeared on freshly excised pre-cancerous tissues from SIST patients, incubated, and eluted with SIF. The fluorescence intensity of the eluate was measured to determine the content of NPs and calculate the bio-adhesion rate. After elution, the mucosal tissues were immediately fixed in formalin, embedded in OCT, frozen sliced (Leica, Wetzlar, Germany), stained with DAPI, observed by CLSM, and quantified the fluorescence intensity by Image J software. In addition, to study whether proper mucus coverage was maintained on ex vivo tissue, we also performed the elution experiment without the exposure of NPs and observed the mucus under a light microscope.

## Bio-retention on GI tract

ICR mice (6 weeks old, weighting 30 ± 2 g) were fasted overnight, oral administrated with fluorescent NPs (2 mg, $n = 3$), and euthanized at preset time intervals. The GI tracts were isolated, subjected to fluorescent imaging using IVIS Lumina Series III Living Image System, and evaluated the mean fluorescence signal intensity of each tissue by ROI analysis.

## Small intestinal villi absorption

The in vivo absorption of NPs by small intestinal villi was studied by CLSM and bio-TEM. Briefly, normal saline and the normal saline suspension of RITC labeled NPs (100 mg/kg, $n = 3$) were respectively oral administrated to mice, which were fasted overnight with free access to water. At the preset time points, the mice were euthanized and the small intestines were isolated, rinsed with SIF, fixed in formalin, embedded in OCT, frozen sliced, stained with DAPI, observed by CLSM and quantified the fluorescence intensity by Image J software. For a competitive adsorption studies, NPs were labeled with different fluorescein and co-administrated to mice (50 mg/kg for each NPs, $n = 3$), and the intestinal villi was observed at 2 h post administration.

Besides, NPs (100 mg/kg, $n = 3$) were oral administrated to mice and executed at 2, 4 and 24 h after administration. Small sections of the duodenum, jejunum and ileum samples were removed and treated by the standard method for bio-TEM observation. Meanwhile, the whole stomach and small intestine were separated, placed in formalin, embedded in paraffin, sectioned, dewaxed, and stained with HE for histological studies.

To further detect the oral adsorption ability and mechanism, the uptake of NPs was performed on ex vivo small intestine of animal and human. In a typical run, the fresh excised intestinal segments of rat (5 cm) and human (5 cm, tumor-adjacent tissues) were collected, washed, ligated both side, injected with SIF (2 ml and 10 ml for rat and human intestinal segments, respectively), pre-incubated for 30 min in

Tyrode's solution at 37 °C for 15 min, and replaced with RITC labeled NPs suspended in SIF (1.5 mg/ml, 2 ml and 10 ml for rat and human intestinal segments, respectively, $n = 3$). After incubation in Tyrode's solution at 37 °C for 1.5 h, the intestinal segments were untied from ligations, rinsed with SIF, fixed in formalin, part of the tissues were embedded in OCT, frozen sliced, stained with DAPI, and observed by CLSM, the other tissues were embedded in paraffin, sectioned, dewaxed, and stained with HE for histological studies. Besides, the fresh excised intestinal segments were pre-treated with the SIF solution of P-gp inhibitor VER (1 mmol/l) for 30 min before incubation with RITC labeled NPs.

### Oral adsorption and bio-distribution

To quantitative study the oral adsorption and distribution of NPs, mice were oral administrated with SSN, MSN, VSN and CVSN (30 mg/kg, $n = 3$), respectively. At the predesigned time intervals, blood samples (80 µl) were taken and centrifuged to collect plasma, digested using perchloric acid, and diluted with HCl (3%). Meanwhile, mice were oral administrated with NPs (30 mg/kg) and were sacrificed at 24 h or 7 days ($n = 3$, respectively) post administration. Their main organs, including heart, liver, spleen, lung, kidney and brain, were removed, weighted, homogenized, and digested using nitric acid. The Si contents in the plasma and organ samples were detected by ICP-MS (iCAP RQ, Thermo Scientific Co. Ltd., USA) using the calibration curve.

Moreover, mice were fasted overnight, oral administrated with RITC labeled NPs (2 mg, $n = 3$), and euthanized at preset time intervals. The major organs were isolated and subjected to fluorescent imaging using IVIS Lumina Series III Living Image System, and evaluated the mean fluorescence signal intensity of each organ by ROI analysis.

### Cellular uptake

Caco-2 cells in logarithmic growth phase were seeded in 24-well plates at a density of $5 \times 10^4$ cells/well, in which 1 ml complete medium was added and placed in a 37 °C incubator with 5% $CO_2$ for 48 h. When the cells grew to 80–90% in the pore plates, the culture medium was discarded and replaced with 1 ml RITC labeled NPs (0.25 mg/ml, $n = 3$), respectively. After incubation at 37 °C for 4 h, the cells were washed three times with cold PBS, fixed in 4% formaldehyde and stained by DAPI for 15 min. Afterward, cells were washed twice with PBS in ice bath and sealed the climbing tablets with 90% glycerin solution. The uptake of NPs was analyzed by CLSM and quantified the fluorescence intensity by Image J software.

$$Cellurar\ Uptake\ Rate\,(\%) = \frac{I_R}{I_D} \times 100\% \qquad (3)$$

Where $I_R$ and $I_D$ represent the fluorescence intensity of RITC and DAPI, respectively[48]. Furthermore, a portion of cells were also treated with trypsin, harvested by centrifugation and quantified by FCM (CytoFLEX, Beckman coulter, USA). To illustrate the cell uptake mechanisms, cells were pre-incubated with specific endocytic inhibitors including CHL (10 µg/ml), CYT (5 µg/ml) and Mβ-CD (1 µg/ml) for 30 min at 37 °C. Then cells were incubated with RITC labeled NPs for another 4 h, and the cellular uptake was analyzed by CLSM and FCM, respectively.

In addition, to observe the cellular uptake behavior, cells were incubated with NPs (0.25 mg/ml, $n = 3$) for 4 h and collected for bio-TEM observation.

### Drug loading and release

To illustrate the pharmaceutical properties of NPs as Nano-DDS, IMC was selected as the model drug and was loaded into MSN, VSN, and CVSN exhibiting nanopores according to the solvent evaporation method. For the preparation of IMC@MSN, 60 mg MSN was added into 2 ml IMC acetone solution (10 mg/ml) in a light-proof vial at the drug/carrier ratio of 1:3 (w/w), sealed and stirred for 24 h. Then, the

samples were dried in a vacuum oven to evaporate the solvent and obtain IMC@MSN. IMC@VSN and IMC@CVSN were synthesized through the same procedure, except that VSN and CVSN were used instead of MSN. To measure the drug loading capacity, 3 mg of drug loaded samples were respectively dispersed in 50 ml methanol. After ultrasonication for 1 h, IMC was completely extracted from the nanopores of the NPs, filtered, and measured the concentration by UV-vis at 320 nm.

With the aim of studying the interactions between IMC and NPs, FTIR and nitrogen adsorption/desorption tests of IMC@MSN, IMC@VSN and IMC@CVSN were carried out. NPs before and after drug loading were also subjected to XRD and DSC analysis on an X-ray diffractometer (EMPYREAN, PANalytical, Netherlands) at the diffraction angles (2θ) of 5–40°, and a microcomputer differential thermal analysis instrument (Q1000, TA Instrument, USA) from 30–250 °C, respectively. The wettability of IMC loaded NPs were also evaluated and compared with IMC.

The release of IMC from NPs was investigated using a dissolution device (ZRS-8G, Huanghai Pharmaceutical, China) via the USP II paddle method at 37 °C and 50 rpm. IMC, IMC@MSN, IMC@VSN, and IMC@CVSN (containing 10 mg IMC) were precisely weighed and put into 250 ml dissolution mediums (SGF, pH 6.5 PBS, SIF, and SBF). 3 ml aliquots were removed at pre-set time points and supplemented with 3 ml fresh dissolution medium to keep a constant volume. The samples were filtered and sent to UV-vis analysis at 318 nm.

### Ex vivo intestinal absorption

Everted intestinal sacs model was employed to study the intestinal transport of drug. The fresh excised small intestine of rat was cleaned, cut into 8 cm segments, turned the mucosal side out, ligated at both ends, injected with 2 ml Tyrode's solution, and immersed in 15 ml the Tyrode's solution of IMC and IMC loaded NPs (containing 5 mg IMC, $n = 3$) at 37 °C and the atmosphere of 95% $O_2$ and 5% $CO_2$. At the present time points, 100 µl solution was collected from the inside serosal side and added with 100 µl Tyrode's solution. Then the samples were centrifuged and measured the ultraviolet absorption via a microplate reader at 320 nm, and calculated the drug content according to the standard curve.

### In vivo pharmacokinetics study

To study the bioavailability of IMC loaded NPs, rats were randomly divided into 4 groups ($n = 3$). IMC and IMC loaded NPs (at a dose equivalent to 35 mg/kg IMC) were oral administrated to rats following fasting overnight. Blood samples were collected from the retro-orbital venous sinus at the pre-set time intervals, centrifuged to separate the plasma, and stored at −20 °C for the further high performance liquid chromatography (HPLC) analysis. 200 µl plasma were pretreated with 80 µl internal standard solution (nimesulide, 400 mg/ml), 20 ml methanol, and 400 ml acetonitrile. The system was vortexed for 2 min, centrifuged at 6804 g for 5 min and then injected (100 µl) into a HPLC device to measure the IMC concentration. The separation was executed on a Diamonsil C18 column (Ser. No. 8034287, 5 µm, 250 × 4.6 mm) with a C18 pre-column; the mobile phase was methanol/ pH 7.3 PBS (7:3, v/v) with a flow rate of 1 ml/min; the UV detector was performed for content determination at 320 nm; the temperature of the column was set at 30 °C; the main pharmacokinetic parameters were obtained using DAS 2.0 software.

### Pharmacodynamics study

The anti-inflammatory effects of IMC loaded NPs were evaluated by MAST, MEST, and MWT[49–51]. First, rats were randomly divided into 5 groups ($n = 3$), including normal saline (negative group), IMC (positive group), IMC@MSN, IMC@VSN, and IMC@CVSN groups, and were oral treated with normal saline and the normal saline suspension of IMC, IMC@MSN, IMC@VSN, and IMC@CVSN (equivalent to 35 mg/kg),

respectively, after overnight fasting. At 30 min post administration, 0.1 ml λ-carrageenan solution (1%, w/v) was injected into the right hind footpad of rats to induce acute inflammation intumesce of the paw. The perimeters of rat right hind paw ankle were recorded at the initial time and the desired time, and calculated the swelling rate and repression rate using the following equations:

$$Swelling\ Rate\,(\%) = \frac{c_t - c_0}{c_0} \times 100\% \qquad (4)$$

$$Repression\ Rate\,(\%) = \frac{(c_t - c_0)_{negative\ control} - (c_t - c_0)_{test}}{(c_t - c_0)_{negative\ control} - (c_t - c_0)_{positive\ control}} \times 100\% \qquad (5)$$

Where $c_0$ and $c_t$ stand for the circumferences of ankles measured at the initiation time and test time, respectively. After the measurement at the final timepoint (6 h), the rats were immediately executed. Their right hind feet were cut and photographed, and the inflammation tissues were isolated, fixed in formalin, embedded in paraffin, sectioned, stained with HE, and observed under a light microscope. The damage score was appraised according the severity of tissue injury on a 0–4 scale: 0–almost normal tissue, 1–one or two lesions, 2–severe lesions, 3–very severe lesions, and 4–tissue full of lesions.

Furthermore, to evaluate the capacity of NPs in reducing the gastrointestinal side effect of IMC, the rats were sacrificed 6 h after oral administration, and the stomach and intestine tissues were immediately removed and sent to histological examination.

Next, the anti-inflammatory effect was further investigated by MEST. 15 mice were randomly divided into 5 groups ($n = 3$), including normal saline (negative group), IMC (positive group), IMC@MSN, IMC@VSN and IMC@CVSN, and were oral treated with normal saline and the normal saline suspension of IMC, IMC@MSN, IMC@VSN and IMC@CVSN (containing 0.6 mg IMC) after fasting overnight. 0.1 ml of xylene was uniformly smeared on the left ear of mice at 30 min after administration to cause edema. After 50 min, the mice were executed and both ears (including the swollen and untreated) were cut off along the contour line, punched out into round pieces by using a 0.6 cm diameter hole punch, weighted and calculated the swelling rate and repression rate according to equation:

$$Swelling\ Rate\,(\%) = \frac{w_L - w_R}{w_R} \times 100\% \qquad (6)$$

$$Repression\ Rate\,(\%) = \frac{(w_L - w_R)_{negative\ control} - (w_L - w_R)_{test}}{(w_L - w_R)_{negative\ control} - (w_L - w_R)_{positive\ control}} \times 100\% \qquad (7)$$

Where $w_L$ and $w_R$ stand for the weights of the pieces cut from the left ear and right ear of the same mice, respectively. After that, the ear tissues were sent to histological examination and damage score assessment.

Finally, the analgesic effect was assessed via MWT. Mice were randomly divided into 5 groups ($n = 3$), including normal saline (negative group), IMC (positive group), IMC@MSN, IMC@VSN and IMC@CVSN ($n = 3$), and were oral treated with normal saline and the normal saline suspension of IMC, IMC@MSN, IMC@VSN and IMC@CVSN (containing 0.6 mg IMC) after fasting overnight. At 1 h after administration, 0.6 ml acetic acid (0.8%, v/v) was injected abdominally to provoke pain, and the mice were placed individually and observed the writhing times within 20 min. The repression rate

was calculated according to the equation:

$$Repression\ Rate\,(\%) = \frac{n_{negative\ control} - n_{test}}{n_{negative\ control} - n_{positive\ control}} \times 100\% \qquad (8)$$

Where $n$ is the number of writhing times.

## Versatility of CVSN as oral Nano-DDS

To illustrate the versatility properties of CVSN as Nano-DDS, a series of NSAIDs, including nimesulide (NMS), acetaminophen (AC), aspirin (ASP), celecoxib (CEL), flurbiprofen (FB), ibuprofen (IBU) and IMC belonging to BCS 2-4 was loaded into CVSN according to the solvent evaporation method at the drug/carrier ratio of 1:3 (w/w). FTIR and XRD studies on these Nano-DDS were conducted to definite the interactions between CVSN and NSAIDs. After that, the ex vivo intestinal transport of NMS@CVSN, AC@CVSN, ASP@CVSN, CEL@CVSN, FB@CVSN, IBU@CVSN, and IMC@CVSN were investigated via everted intestinal sacs model by measuring the ultraviolet absorption at 395, 257, 235, 250, 247, 265, and 320 nm, respectively, and calculating the drug content according to the standard curves. Finally, the therapeutic effects of the Nano-DDS were measured by MEST model, wherein 0.5 ml of xylene was used to induce serious swelling.

## Statistical and reproducibility

All experiments were performed in triplicate unless otherwise stated. A representative image of three replicates from each group was shown. The results were presented as mean ± standard deviation (SD). Two tailed Student's $t$-test was conducted to compare experimental groups. The differences were considered statistically significant for $P < 0.05$. The levels of significance were set at the probability of *$P < 0.05$, **$P < 0.01$, and ***$P < 0.001$.

## Reporting summary

Further information on research design is available in the Nature Portfolio Reporting Summary linked to this article.

## Data availability

Source data are provided with this paper. Source data are available for Figs. 1b, f, 2, 3b, d, e, g, k, l, 4c, d, f, 5c–f, j–n, 6c, d, f, 7, 8d–o, Supplementary Figs. 2–8, 10–21, 23, 24, 26–32, 34, 35 in the associated "Source Data" file. The source data have been deposited in the Figshare database (10.6084/m9.figshare.24476692). All the other data that support the findings of this study are available within the Article and its Supplementary Information files and from the corresponding author. Source data are provided with this paper.

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

## Acknowledgements

This work was supported by the National Natural Science Foundation of China (No. 81903550 [H.L.]), Natural Science Youth Foundation of Sichuan Province (No. 2023NSFSC1769 [K.G.]), and Central Nervous System Drug Key Laboratory of Sichuan Province (No. 230014-01SZ [K.G.])

## Author contributions

Z.S., L.X. and R.D. contributed equally to this work. X.C. and H.L. conceived and designed the research. H.Y., Y.C. and H.L. supervised the project. Z.S., L.X., R.D., M.W., X.Y. and B.Z. performed the research. X.L., K.G. Y.H. and T.L. analyzed and interpreted the data. X.C., R.D. and Y.H. provided useful suggestions. Z.S. and H.L. wrote the manuscript. Z.S., L.X., R.D. and H.L. revised the manuscript. All authors have given approval to the final version of the manuscript.

## Competing interests

The authors declare no competing interests.
