## [Peer Review File · Nature Communications]

Reviewers' Comments:

Reviewer #1:

Remarks to the Author:

This manuscript reports a drug delivery system of silica nanoparticles modified by chiral molecules that mimic the structure and function of viruses to enhance the efficiency of oral administration. The experimental results showed that compared with smooth and mesoporous silica nanoparticles, chiral virus silica nanoparticles had obvious advantages in terms of digestive tract mucous penetration, gastrointestinal retention, and oral administration efficiency. The manuscript has clear logic and a complete experimental design, but there are some problems in detail. And there are some questions that should be addressed as detailed below. I recommend acceptance of this work only after careful revisions.

1. The author mentioned several times that chiral molecular modification can improve the topological roughness of CVSN, but the concept of topological roughness is not defined or explained in the manuscript.
2. Some TEM and SEM images in Figure 1 need to be retaken. TEM images of VSN and CVSN appear to be similar to the shape of dendritic mesoporous nanoparticles, which are inconsistent with the amplifying images, and the amplifying images are very fuzzy. SEM images are also fuzzy, requiring the selection of more dispersed nanoparticles rather than blocky nanoparticles for characterization, and the contrast also needs to be adjusted to the right value.
3. In Figure 2B, why is the starting point of MSN, VSN, and CVSN, not 100%?
4. On page 5 line 147, the authors state that "MSN, VSN, and CVSN all showed distinct type IV adsorption/desorption isotherms with distinct hysteresis loops, ... demonstrating the absence of pore structure (Fig. 2C)." The pore size distribution curves and TEM images showed that the pore size of the nanoparticles was less than 5 nm, corresponding to the middle/low-pressure part of the nitrogen adsorption and desorption curve, which was not enough to generate hysteresis loop. The hysteresis loop of the nitrogen adsorption and desorption curve in the figure is located in the high-pressure section, which is more likely to be the pores formed by the stacking of nanoparticles. The analysis in Figure 7C has the same problem.
5. In Figure 2M, why is there a big difference in the green fluorescence thickness of the mucus in the Z direction?
6. Chiral recognition can enhance the interaction between nanoparticles and organisms. While enhancing the uptake of nanoparticles by cells, will this interaction in turn inhibit the transport of nanoparticles, that is, the penetration effect of nanoparticles?
7. The fluorescence intensity in Figure 6B appears to be inconsistent with the values described in the manuscript. It is suggested to supplement the flow cytometry data for quantitative analysis. The formula for calculating the endocytosis of nanoparticles by the image fluorescence intensity should also be reflected in the method section.
8. In Figure 6A, why do SSN and MSN correspond to rolling friction and sliding friction respectively?
9. Some minor problems appear in the manuscript:
 - ① A legend should be given for what the green and yellow patterns in Figure 1A represent respectively.
 - ② Figures should be numbered from left to right and from top to bottom. The sequence numbers in Figures 2, 3, 4, and 5 are messy and need to be reconsidered.
 - ③ On page 5 line 138, the authors state that "SSN and MSN were monodisperse mesoporous spheres." SSN is not mesoporous spheres.
 - ④ On page 5 line 150, the authors state that "The results showed the mesoporous structures of NPs, in which the pore size of MSN, VSN and CVSN were 3.2, 2.9 and 2.5 nm, respectively, due to the presence of nanospikes and the modification of L-Ala." The synthesis methods of MSN and VSN are completely different, the epitaxial growth of VSN is not the cause of small pore size, so the causality in the sentence does not exist.
 - ⑤ The spelling of the peristaltic pump in Figure 3A is wrong.
 - ⑥ On page 16 line 383, the authors state that "In comparison, the pretreatment of CHL and M β -CD resulted in a 60.07% and 74.31% ... while the use of CYT showed no significant (less than 15%) inhibitory effect." M β -CD should be replaced by CYT.
 - ⑦ In this manuscript, CTAB was removed by calcination, which would make the dispersion of nanoparticles worse. In the subsequent study, it is better to remove the surfactant by extraction.

Reviewer #2:

Remarks to the Author:

This manuscript outlines the development of an innovative, virus-mimicking nanoparticle to tune interactions with the intestinal mucosa for improved bioavailability of orally delivered drugs. The authors provide an impressive and comprehensive set of materials characterization data, followed by biological assays in vitro and in vivo studying oral delivery of the small molecule indomethacin using these nanoparticles. The arguments presented in the results are well-structured, but the language should be carefully reviewed to ensure clarity of the point made and conclusions drawn. Additionally, the authors should consider whether the massive amount of data collected for this work would be more clearly presented split into two separate manuscripts, as discussed below in point 8. To that end, there are a number of points that the authors should consider to strengthen their manuscript:

1. Lines 71-93 invoke relevance of the COVID-19 pandemic to this work, then list the results of approximately 12 previous studies without describing how these relate to or inform the current work. Instead of trying to fit in the pandemic buzz and a miniature literature review, the authors should instead focus on providing background information that more clearly explains why VSNS and CVSNS are the right choice for this drug delivery system.
2. In the motivation for and discussion of Figure 1, the authors should provide more context for why they are using L-alanine functionalization, and clearer description the process. Citations must be provided for the assertion that surface chirality dictates mucosal interactions. AVSN should be defined in the text, with more information on the synthesis steps that separate MSNs, VSNS, AVSNs, and CVSNS. FTIR is generally considered a qualitative metric for the presence of different functional groups, but a qualitative measure of the surface coverage / number of L-alanine per NP / something similar would be helpful for understanding the system. Additionally, the authors eventually conclude that chirality of surface is critical for the system, but there is very little separation of data between VSNS and CVSNS to support that. Would addition of a chiral control – such as D-Alanine – provide better evidence for these conclusions?
3. Throughout the manuscript, it would be helpful to include more descriptive figure captions, and imperative to provide larger labels in figure panels as the current figures are often illegible. There are additionally some supplementary figures where the y axes are not labelled. Finally, some figures use the same color scheme for the set of 4 NP materials, but the identity linked between each color changes between panels – these should be standardized.
4. The 3D orientation of Figure 2M makes it difficult for the reader to understand where nanoparticles are localizing. Orthogonal projections to a 2D plane may be better to represent this data. Additionally, the red/green color scheme is not friendly to readers with color blindness, and the authors should consider using a different one.
5. For the adhesion assay outlined in Figure 3, it has generally been difficult for the field to maintain proper mucus coverage on ex vivo tissue, and the authors do not provide any evidence that theirs is intact. Either characterization of this model, or citations to previous work that did properly characterize it, should be provided to ensure that particles are adhering to biologically relevant tissue and not deprotected epithelial cells. In addition, Figure 3 should be updated to more clearly show which panels correspond to mouse intestines and which to human intestines.
6. For the discussion of Figure 5, the authors should provide more information on how Rhodamine is incorporated into the nanoparticles, as well as how this could affect experimental results. For example, is rhodamine able to leach back out of the particles? If so, the author's conclusions about efflux of SSNS and about partition coefficients both more likely reflect the behavior of free rhodamine. The authors should also discuss, with sources, the implications of their findings of silica accumulating in heart tissue, as well as include information on the time points for histological analysis.

7. The mechanism presented in Figure 6A is done so without evidence, and is never directly supported in the text. The authors must properly justify these claims. It should also be noted that Supplementary Figure Panels 22C through 22G are never called from the main text.

8. The authors mention poor solubility of indomethacin, but otherwise do not justify its use as a model drug, which raises concerns given the 100% bioavailability of orally administered indomethacin (<https://pubmed.ncbi.nlm.nih.gov/26865183/>). The drug release and particle uptake data presented better support the idea that the Nano-DDS developed here are releasing indomethacin into the intestinal lumen, and that the free small molecule is diffusing on its own across the epithelial barrier. To this end, the authors should consider splitting the massive amount of data that they have generated into two separate manuscripts: (1) using MSNs or VSNs to deliver epithelium-permeable drugs like indomethacin into the intestines for rapid adsorption, and (2) chiral functionalization of VSNs to CVSNs to drive nanoparticle uptake across the epithelium and deliver drugs that are not able to cross the barrier on their own.

9. For the discussion of Figure 8, the authors should consider adding a pathologists' report with quantitative metrics to support the conclusions that they draw from histology images. The arguments would be better supported with some literature citations justifying pre-treatment of animals in the inflammation models, as well as establishing the two models employed. The figure itself would also benefit from better delineation between panels – which correspond to ankle swelling vs. ear?

10. The authors should make sure that their conclusions do not overreach the work shown. In line 520, it is unclear from the manuscript what is meant by "signal transfer at the bio-interface." Similarly, line 532 mentions "low cost, high stability, ease of large-scale production" that are not discussed in the main text.

Dear editors and reviewers,

On behalf of my co-authors, we are truly grateful for giving us an opportunity to revise our manuscript, we appreciate editors and reviewers very much for their positive and constructive comments and suggestions on our manuscript entitled “**Nanoparticles Exhibiting Virus-Mimic Surface Topology for Enhanced Oral Delivery**” (Journal: **Nature Communications**; No. NCOMMS-23-23385). Those comments are all valuable and very helpful for revising and improving our paper, as well as the important guiding significance to our researches. We have studied comments carefully and have made correction which we hope meet with approval. Revised contents were marked in blue in the revised manuscript.

The point-by-point answers to the comments and suggestions were listed as below.

Reviewer #1:

This manuscript reports a drug delivery system of silica nanoparticles modified by chiral molecules that mimic the structure and function of viruses to enhance the efficiency of oral administration. The experimental results showed that compared with smooth and mesoporous silica nanoparticles, chiral virus silica nanoparticles had obvious advantages in terms of digestive tract mucous penetration, gastrointestinal retention, and oral administration efficiency. The manuscript has clear logic and a complete experimental design, but there are some problems in detail. And there are some questions that should be addressed as detailed below. I recommend acceptance of this work only after careful revisions.

1. The author mentioned several times that chiral molecular modification can improve the topological roughness of CVSN, but the concept of topological roughness is not defined or explained in the manuscript.

Response: Many thanks for your kind work. These suggestions are valuable and very helpful to improve the quality of our manuscript, as well as guide our research in the near future. According to your advice, we thoughtful revised the manuscript and made some major changes in the manuscript.

1) Firstly, considering your suggestions on quantitative analysis for cellular uptake and penetration effect of NPs, we had supplied the flow cytometry (FCM) data of NPs, wherein the internalization efficiency ranked as CVSN > VSN > MSN > SSN and was in accordance with the confocal laser scanning microscopy (CLSM) results (Fig. 6d and Fig. S25). After per-treatment with inhibitors, the variation tendency on cellular uptake measured by FCM was basically consistent with the semi-quantitative results acquired by CLSM (Fig. 6f). Meanwhile,

we also performed bio-TEM to directly observe the uptake and transport of NPs, wherein spherical NPs (SSN and MSN) showed poor adhesion on the cell membrane with single contact point and followed by the limited endocytosis into the cells, while VSN and CVSN with virus-mimic surface topology provided multiple contact sites and directly anchored on the cell membrane to facilitate cellular uptake (Fig. 6a). Excitingly, numerous CVSN appeared in the cells.

2) Secondly, to confirm the versatility of CVSN as oral Nano-DDS, we chose a series of nonsteroidal anti-inflammatory drugs (NSAIDs) with problems of oral adsorption (including nimesulide [NMS], acetaminophen [AC], aspirin [ASP], celecoxib [CEL], flurbiprofen [FB], ibuprofen [IBU] and IMC) belonging to biopharmaceutical classification system (BCS) 2-4, characterized by poor solubility and/or low permeability, and encapsulated them into CVSN to construct Nano-DDS (Fig. 8a–b and Table S5). We confirmed the successfully loading and high dispersion of drugs in amorphous state via FTIR and XRD (Fig. S34), and investigated the *ex vivo* intestinal transport and *in vivo* bioavailability of these Nano-DDS via everted intestinal sacs model (Fig. 8a) and mouse ear swelling test model (MEST, Fig. 8m), respectively. The results showed that CVSN was able to improve the oral adsorption of almost all these NSAIDs with diverse structures, molecular weights and functional groups, wherein the transport rate raised about 1.60-7.71 times (Fig. 8c–l) and the anti-inflammatory effect improved about 1.21-2.29 times, respectively (Fig. 8n–p).

3) Thirdly, other *in vivo* studies were supplied to support the conclusion, including the competitive adhesion study of NPs on the human intestinal mucosa, the bio-adhesion and intestinal permeability of DVSN (VSN modified with D-Ala). The results suggested that, NPs with virus-like shape as well as L-chiral conformation always defeated the counterparts and showed higher retention penetration and absorption on the small intestine (Fig. S15). Before analysis, we treated the intestinal tissues under the storage condition and the experimental procedure, and confirmed proper mucus coverage on the *ex vivo* mucosa (Fig. S12). The *ex vivo* intestinal transport of IMC@NPs were also studied via everted intestinal sacs model, and a strongest intestinal transport was observed for IMC@CVSN, wherein the transport amount of IMC@CVSN was up to 7.71-times than that of IMC (Fig. S31).

4) Fourthly, in the material synthesis part, we had revised the schematic diagram and structural formula of NPs to give more information about the material synthesis (Fig. 1a and Fig. S1), redone the TEM, SEM and some characterizations (Fig. 1c–d, Fig. 2b and Fig. S2), and improved the result descriptions based on your professional advice. We also added the synthesis and characterization of the intermediate product AVSN (amination VSN) for comparison (Fig. 1 and Fig. 2a–e). Furthermore, we had provided more information on the mechanism and method of grafting the fluorescent dyes onto the NPs (Fig. S9–10).

5) Lastly, the structure of the article was adjusted and some results were moved to supporting information to increase the readability and logicity of the article. The quality of all images and tables were improved, in which the images, labels, arrows and font size were amplified for

assessment of accuracy, more descriptive figure captions were supplied, and the figure numbers were rearranged and presented in consecutive order. The entire text was checked under the assistance of ACS Language Editing Services (Serial number: INQ-2151483922_HEHEWC-3). More appropriate references were cited.

6) In addition, as the review point out, after careful consideration, we found the words “topological roughness” was inaccurate, so we replaced the words with “surface roughness” and “surface topologies”, which referred to “the shape, texture, elasticity and roughness of the surface, and the density and spatial orientation of the functional groups” and defined in the introduction. Please see the sentences marked in blue in Page 3.

2. *Some TEM and SEM images in Figure 1 need to be retaken. TEM images of VSN and CVSN appear to be similar to the shape of dendritic mesoporous nanoparticles, which are inconsistent with the amplifying images, and the amplifying images are very fuzzy. SEM images are also fuzzy, requiring the selection of more dispersed nanoparticles rather than blocky nanoparticles for characterization, and the contrast also needs to be adjusted to the right value.*

Response: Thanks very much for your nice suggestions. According to your advice, we had redone the SEM and TEM studies to a higher resolution, and adjusted the images to the right value for comparison. Please see the revised Fig. 1c–d. As presented in the TEM and SEM images, SSN and MSN were uniformed nanospheres, while VSN, AVSN and CVSN were virus-like particles (nanospheres with a large number of nanotubes distributed around their surfaces). All NPs were homogeneous in shape and size, with the average particle size of 80 nm (Fig. 1e–f and Fig. S2). Due to the rough surface and big surface energy, it seemed slightly aggregated. Actually, on account of the small particle size of NPs (~80 nm), we had reached the highest performance of the instruments. So, we also conducted the structural analysis of NPs combined with many other characterization approaches (e.g. DLS, FTIR, N₂ adsorption–desorption tests, SAXS, CD, XRD, AFM, TG and contact angle measurement), and noticed the difference between NPs in surface topologies. Once again, we appreciated for reviewers’ warm work earnestly, and hope that the correction would meet with approval.

3. *In Figure 2B, why is the starting point of MSN, VSN, and CVSN, not 100%?*

Response: Thanks for your careful checks. The starting point of MSN, VSN, and CVSN was not 100% as a consequence of the loose of adsorbed water and the unstable quality state at the start of measurement. Considering your concern, we had reserved the samples in a desiccator and redone the TG study after the stabilization of the instrument, and the starting point of NPs kept at 100% at this time. Please see the revised Fig. 2b. The weight loss (from 25 to 700°C) of SSN, MSN, VSN, AVSN and CVSN were 0.91%, 2.51%, 7.33%, 14.32% and 22.07%, respectively, demonstrating the thermostability of silica framework and the successful grafting of functional groups. CVSN experienced intense weight loss on account of the organic chiral

molecule modification, which was measured to be 14.75%.

4. *On page 5 line 147, the authors state that “MSN, VSN, and CVSN all showed distinct type IV adsorption/desorption isotherms with distinct hysteresis loops, ... demonstrating the absence of pore structure (Fig. 2C).” The pore size distribution curves and TEM images showed that the pore size of the nanoparticles was less than 5 nm, corresponding to the middle/low-pressure part of the nitrogen adsorption and desorption curve, which was not enough to generate hysteresis loop. The hysteresis loop of the nitrogen absorption and desorption curve in the figure is located in the high-pressure section, which is more likely to be the pores formed by the stacking of nanoparticles. The analysis in Figure 7C has the same problem.*

Response: Many thanks for your professional suggestion. We had corrected the descriptions in the manuscript to “Besides, MSN, VSN, AVSN and CVSN all showed distinct type IV adsorption/desorption isotherms with hysteresis loops in the high-pressure section, most likely to be the pores formed by the stacking of NPs” Meanwhile, the same statement in Fig. 7C (had revised to Fig. S30) as “as evidenced by the rearward shift of hysteresis loops” was deleted. Please see the sentences marked in blue in Page 5 and Page 19.

5. *In Figure 2M, why is there a big difference in the green fluorescence thickness of the mucus in the Z direction?*

Response: Thanks very much for your comments. The experiment was carried out *ex vivo* on the small intestines of SD rats, wherein we first marked the mucus by 500 μ l FITC and then incubated with RITC labeled NPs. During this procedure, we performed standardized operations, and I think the difference in the green fluorescence thickness of the mucus was because of the individual difference of animals. Based on your advice, we had replaced the image with a parallel data, wherein the thickness of the mucus was similar to the other groups. Please see the revised Fig. 2m. Meanwhile, according to the advice of reviewer 2, we had supplied the 2D plane orientation pictures of NPs permeating through the mucus, which helped to locate the penetration depth and relative location of NPs. The red/green color scheme was changed to a red/blue color scheme to improve the contrast ratio as well as to be friendly to the readers with color blindness. As can be seen from the 3D images, only weak fluorescent signals from SSN and MSN were distributed in the mucus on the proximal luminal side of the intestine, whereas potent fluorescent signals from VSN and CVSN were observed in the entire mucus layer with good extensibility (Fig. 2m–o). Particularly, CVSN could effectively diffuse into the deepest and widest sections of the mucus layer, evidenced by co-localization of fluorescence signals in the 2D plane orientation pictures (Fig. 2m).

6. *Chiral recognition can enhance the interaction between nanoparticles and organisms. While enhancing the uptake of nanoparticles by cells, will this interaction in turn inhibit the*

transport of nanoparticles, that is, the penetration effect of nanoparticles?

Response: Thanks for your kind work. In this study, we first demonstrated that CVSNS exhibiting the marriage of virus-mimic spikey surface and molecular chirality produced positive effect on the adhesion ability on bio-membrane systems (including the macroscopic intestinal mucosa and the microscopic cell membrane). In both rat and human intestinal tissues and the cell membrane of Caco-2 cells, the bio-adhesion ability of NPs followed the order of CVSNS > VSN > MSN > SSN (Fig. 3, Fig.S13 and 15, and Fig. 6b–c).

To illustrate whether enhancing the adhesion and uptake of NPs would facilitate or in turn inhibit their transport ability, we investigated the penetration effect of NPs through intestinal permeability study, *in vivo* absorption and distribution evaluations and cellular internalization study. i) First, we monitored the adsorption of NPs through the intestinal villi of mice (*in vivo*) rat and human (*ex vivo*), and found the penetration and transport of CVSNS on the small intestine was most efficient (Fig. 4a–e, Fig. 5b–f and i–k, Fig. S17 and Fig.S19–21). ii) Then we directly determined the oral bioavailability of NPs by measuring the Si element in the blood via ICP-MS. According to Fig. 5n, CVSNS could rapidly adsorb in the blood at 4 h after administration, and achieved satisfactory oral bioavailability with the largest area under the curve (AUC) and efficiently distribute in almost all organs (Fig. 5l–m). iii) Finally, we assessed the transport of NPs on cellular level via CLSM, FCM and bio-TEM. CLSM images discovered CVSNS displayed strong fluorescent signals locating near the nucleus or intra nucleus, implying active intracellular transport, which was in accordance with the FCM and bio-TEM findings that significant number of CVSNS was fast entering into the cells but not just adhering on the cell membrane (Fig. 6a–d).

From this knowable, we believed that the virus-mimic strategy of NPs showed both enhanced adhesion and penetration effect.

7. *The fluorescence intensity in Figure 6B appears to be inconsistent with the values described in the manuscript. It is suggested to supplement the flow cytometry data for quantitative analysis. The formula for calculating the endocytosis of nanoparticles by the image fluorescence intensity should also be reflected in the method section.*

Response: We sincerely appreciate the valuable comments. 1) First, as the reviewer points out, we had a clerical error in the manuscript (Mβ-CD should be replaced by CYT), which caused the inconsistency with the figures. In the revised manuscript, we had corrected the clerical error, revised the inconsistent data and improved the description. Please see the sentences marked in blue in Page 16. By the way, we had added the formula for calculating the cellular uptake rate by fluorescence intensity in the method section.

$$\text{Cellular Uptake Rate} = \frac{I_R}{I_D} \times 100\%$$

Where IR and ID represent fluorescence intensity of RITC and fluorescence intensity of DAPI,

respectively. Please see the sentences marked in blue in Page 28.

2) Secondly, according to your advice, we had supplied the FCM data for quantitative analysis. The FCM results was basically consistent with the tendency in CLSM study, wherein the internalization efficiency as well as the fluorescence intensity ranked as CVSN > VSN > MSN > SSN (Fig. 6d and Fig. S25). After pretreated with CHL, CYT and M β -CD, the inhibition rate measured to be 26.68%, 37.57% and 74.60% for SSN, 26.36%, 71.72% and 57.48% for MSN, 56.50%, 79.31% and 8.15% for VSN and 17.62%, 66.81% and 60.91% for CVSN, showing good agreement with the semi-quantitative data of CLSM (Fig. 6f and Fig. S25).

3) Thirdly, to visually confirm the cellular uptake behavior, the cells were incubated with NPs for 4 h and sent to bio-TEM observation. Herein, spherical NPs (SSN and MSN) showed poor adhesion on the cell membrane with a single contact point and followed by the limited endocytosis into the cells (Fig. 6a). Conversely, VSN and CVSN with virus-mimic surface topology provided multiple contact sites and directly anchored on the cell membrane to facilitate cellular uptake. Excitingly, numerous CVSN appeared in the cells.

Taken together, based on the bio-TEM, CLSM and FCM results, we confirmed the superior internalization efficiency of CVSN.

8. *In Figure 6A, why do SSN and MSN correspond to rolling friction and sliding fraction respectively?*

Response: We sincerely appreciate the valuable comments. In the experimental conception phase, we designed NPs exhibiting virus-mimic surface topology and aimed to explore its structure–function relationships on biological effect by comparing with NPs exhibiting different surface roughness (including the smooth SSN, mesoporous MSN and virus-like VSN). In our assumption, spherical SSN had smooth surface, which was associated with small friction coefficient and good mobility, and was hard to provide a fixed friction surface. For MSN, VSN and CVSN, the surface roughness increased in turn. The pore structure as well as the spiky surface of NPs was able to provide definite contact points, providing material basics for sliding friction. Once contacting with the biological membranes, MSN, VSN and CVSN were theoretically related to the large frictional forces comparing to SSN.

However, due to technological limitation, the assumption had not been directly verified, although many studies indicated the improved adhesion and penetration effect of CVSN both *in vitro* and *in vivo*. Considering your advice as well as the similar suggestion proposed by review 2 (conclusions do not overreach the work shown), we deleted the relative descriptions about rolling friction and sliding fraction after carefully consideration, and remained the descriptions on “single-site contact” and “multi-site contact”, which was directly observed by bio-TEM. Once again, thanks for your constructive suggestions.

9. *Some minor problems appear in the manuscript:*

① *A legend should be given for what the green and yellow patterns in Figure 1A represent respectively.*

Response: Thank you very much for your suggestions. We had supplied the legend for what the green and yellow patterns in Figure 1a. Meanwhile, we also added the molecular formula for the synthesis of NPs. Please see the revised Fig. 1a and Fig. S1.

② *Figures should be numbered from left to right and from top to bottom. The sequence numbers in Figures 2, 3, 4, and 5 are messy and need to be reconsidered.*

Response: Thank you very much for your suggestions. The quality of all images and tables were improved, in which the images, labels, arrows and font size were amplified for assessment of accuracy, more descriptive figure captions were supplied, and the figure numbers were rearranged and presented in consecutive order. The sequence numbers in Figures 2, 3, 4, and 5 are reconsidered and numbered. However, in Figures 4 and 5, the left and right part involved to different experiments, so we tried our best to number the images from left to right and from top to bottom in two parts, increased the spacing between the left and right parts, and added more descriptive figure captions. Please see the revised Fig. 2–5. We had tried our best for typesetting, and hope the change could meet the approval.

③ *On page 5 line 138, the authors state that “SSN and MSN were monodisperse mesoporous spheres.” SSN is not mesoporous spheres.*

Response: Thank you very much for your suggestions. The statement had revised to SSN and MSN were monodisperse nanospheres. Please see the sentence marked in blue in Page 5.

④ *On page 5 line 150, the authors state that “The results showed the mesoporous structures of NPs, in which the pore size of MSN, VSN and CVSN were 3.2, 2.9 and 2.5 nm, respectively, due to the presence of nanopikes and the modification of L-Ala.” The synthesis methods of MSN and VSN are completely different, the epitaxial growth of VSN is not the cause of small pore size, so the causality in the sentence does not exist.*

Response: Thanks for your kindly remind. According to your advice, the unprecise expression was deleted and the sentence was revised to “The pore size distribution curve and the calculated texture parameters of NPs were summarized in Fig. 2e and Table 1, respectively, in which the pore size of MSN, VSN and CVSN were 3.2, 2.9 and 2.5 nm, respectively.” Please see the sentence marked in blue in Page 5.

⑤ *The spelling of the peristaltic pump in Figure 3A is wrong.*

Response: Thanks for your nice suggestion. The clerical mistake “peristatic pump” had been corrected to “peristaltic pump”. Please see the revised Fig. 3a and Fig. S15.

⑥ *On page 16 line 383, the authors state that “In comparison, the pretreatment of CHL and Mβ-CD resulted in a 60.07% and 74.31% ... while the use of CYT showed no significant (less*

than 15%) inhibitory effect.” M β -CD should be replaced by CYT.

Response: Thanks very much for your kind work, we felt sorry for our clerical error for M β -CD, which also caused the inconsistency between the fluorescence intensity in Fig. 6b–c and the values described in the manuscript. And M β -CD here had been replaced by CYT in the revised manuscript. Please see the sentence marked in blue in Page 16.

⑦ *In this manuscript, CTAB was removed by calcination, which would make the dispersion of nanoparticles worse. In the subsequent study, it is better to remove the surfactant by extraction.*

Response: Many thanks for your professional opinion on material synthesis, which are valuable and very helpful to improve the quality of our manuscript, as well as guide our research in the near future. Based on your advice, we had tried to remove the template by the reflux of HCl/methanol. And we would further improve the synthetic method in the subsequent study. Once again, thanks for your kind work.

Reviewer #2

This manuscript outlines the development of an innovative, virus-mimicking nanoparticle to tune interactions with the intestinal mucosa for improved bioavailability of orally delivered drugs. The authors provide an impressive and comprehensive set of materials characterization data, followed by biological assays in vitro and in vivo studying oral delivery of the small molecule indomethacin using these nanoparticles. The arguments presented in the results are well-structured, but the language should be carefully reviewed to ensure clarity of the point made and conclusions drawn. Additionally, the authors should consider whether the massive amount of data collected for this work would be more clearly presented split into two separate manuscripts, as discussed below in point 8. To that end, there are a number of points that the authors should consider to strengthen their manuscript:

1. *Lines 71-93 invoke relevance of the COVID-19 pandemic to this work, then list the results of approximately 12 previous studies without describing how these relate to or inform the current work. Instead of trying to fit in the pandemic buzz and a miniature literature review, the authors should instead focus on providing background information that more clearly explains why VSNs and CVSNs are the right choice for this drug delivery system.*

Response: Thanks for your nice suggestion. We had studied reviewer’s comments carefully to revise our manuscript and made some major changes in the manuscript.

1) Firstly, considering your suggestion on the logicity of the manuscript by choosing

insoluble drug indomethacin (IMC) as model drug, as well as to verify the versatility of CVSNS as oral Nano-DDS, we further selected a series of NSAIDs with problems of oral adsorption (including nimesulide [NMS], acetaminophen [AC], aspirin [ASP], celecoxib [CEL], flurbiprofen [FB], ibuprofen [IBU] and IMC) belonging to BCS 2–4, characterized by poor solubility and/or low permeability as model drugs, encapsulated them into CVSNS to construct Nano-DDS, and rapidly investigated the *ex vivo* intestinal transport and *in vivo* bioavailability of these Nano-DDS via everted intestinal sacs model and MEST model, respectively. The results indicated that CVSNS was able to improve the oral adsorption of all these NSAIDs with diverse structures, molecular weights, functional groups, solubility and permeability (Fig. 8, Fig. S34 and Table S6). And we would extensively discuss this point below in point 8.

2) Secondly, based on the aforementioned change, the structure of the results section had been improved to 5 parts, including, i) the structure and surface/interface properties of NPs, ii) the bio-adhesion and retention of NPs on the intestinal mucosa, iii) the penetration of NPs through the mucus and epithelium, iv) the oral bioavailability, bio-distribution and biocompatibility of NPs, v) the pharmaceutical properties (drug loading and release properties, pharmacokinetics and pharmacodynamics) of IMC loaded NPs, and vi) the versatility of CVSNS as oral Nano-DDS. The introduction part was simplified, and some results were moved to the supporting information to increase the readability of the article. Under this modification, this work would be more clearly presented with better integrity and logic. Meanwhile, the language was carefully revised to ensure clarity of the point made and conclusions, with the guiding principle of “the conclusions do not overreach the work shown”.

3) Thirdly, for mechanism verification, we directly observed the internalization and transport of NPs into the cells via bio-TEM, in which spherical NPs (SSN and MSN) showed poor adhesion on the cell membrane with single contact point and followed by the limited endocytosis into the cells, while VSN and CVSNS with virus-mimic surface topology provided multiple contact sites and directly anchored on the cell membrane to facilitate cellular uptake (Fig. 6a). For quantitative analysis, we supplied the flow cytometry data of NPs, wherein the internalization efficiency ranked as CVSNS > VSN > MSN > SSN and was in accordance with the CLSM results (Fig. 6d and Fig. S25). Considering your suggestion on chiral recognition, we also prepared DVSNS (VSN modified with D-Ala) and compared the *in vitro* and *in vivo* behaviors between CVSNS and DVSNS via amino acid adsorption, bio-adhesion study, intestinal permeability study and cell internalization efficiency. These results discovered the distinct amino acid behaviors of NPs with different chirality (Fig. 2k–l), and showed superiority of CVSNS on bio-adhesion (Fig. S14), intestinal permeability (Fig. S16), and cellular uptake (Fig. S27), demonstrating that the surface chirality of NPs dictated mucosal interactions via active chiral recognition. Also, the results were consistent with our preliminary work in the chiral recognition and biological effect of NPs exhibiting different molecular chirality, which would be extensively reported below in point 2. In addition, other studies were supplied to further support the superiority of CVSNS, including the competitive adhesion study of NPs on the

human intestinal mucosa (Fig. S15), and the intestinal transport of IMC loaded NPs (Fig. S31). The results indicated that, NPs with virus-like shape as well as L-chiral conformation always defeated the counterparts and showed higher retention, penetration and transport on the small intestine.

4) Lastly, we had revised the schematic diagram and structural formula to give more information about the material synthesis (Fig. 1a and Fig. S1), added the synthesis and characterization of intermediate product AVSN (amination VSN) for comparison (Fig. 1 and Fig. 2a–e) and provided more information on the mechanism and method of grafting the fluorescent dye onto the NPs (Fig. S9–10). The quality of all images and tables were improved, in which the images, labels, arrows and font size were amplified for assessment of accuracy, more descriptive figure captions were supplied, and the figure numbers were rearranged and presented in consecutive order. The entire text was checked under the assistance of ACS Language Editing Services (Serial number: INQ-2151483922_HEHEWC-3). More appropriate references were cited.

5) Particularly, according to your advice on point 1, we had rewritten the relevant part, deleted the relevant expression of COVID-19 pandemic, detected the unsuitable references, and directly proposed the concept of virus bionics with more appropriate references. Please see the sentence marked in blue in Page 2 and Page 23.

We appreciated for reviewers' warm work earnestly, and hope that the corrections would meet with approval.

2. *In the motivation for and discussion of Figure 1, the authors should provide more context for why they are using L-alanine functionalization, and clearer description the process. Citations must be provided for the assertion that surface chirality dictates mucosal interactions. AVSN should be defined in the text, with more information on the synthesis steps that separate MSNs, VSNs, AVSNs, and CVSNs. FTIR is generally considered a qualitative metric for the presence of different functional groups, but a qualitative measure of the surface coverage / number of L-alanine per NP / something similar would be helpful for understanding the system. Additionally, the authors eventually conclude that chirality of surface is critical for the system, but there is very little separation of data between VSNs and CVSNs to support that. Would addition of a chiral control – such as D-Alanine – provide better evidence for these conclusions?*

Response: 1) In the conceptualization phase of this study, we noticed that, as natural creations, all viruses exhibited chiral architectures. That is, the helix structure of DNA or RNA in the core plays a vital role in the gene transcript, translation and expression of viruses, while and the secondary structure of proteins (which are made up of L-type amino acids) is crucial to maintain their spatial stability and bio-responsive activity. In recent years, researchers have discovered that the chirality of nanomaterials also showed significant influences on protein adsorption, cell

adhesion, proliferation and differentiation, cell phagocytosis, cell apoptosis, and disease diagnosis and treatment. Considering that the biological functions of natural assemblies are highly depending on their chirality, we further modified VSN with molecular chiral group (L-Ala) to enable the chiral recognition for functional bionics.

2) In fact, in our preliminary work, we had prepared L/D-Ala modified mesoporous silica nanoparticles (including L/D-Ala modified mesoporous silica nanospheres [L/D-MSN] and L/D-Ala modified mesoporous silica nanorods [L/D-MSR]) to investigate the chirality-mediated biological effect. We found that the introduction of small molecular chiral groups could improve surface/interface properties, increase the bio-adhesion and facilitate the cell uptake as well as improve the intestinal permeability of NPs compared to the achiral NPs. In contrast to corresponding D-chiral NPs, L-MSR presented lower cytotoxicity, higher cellular uptake by Caco-2 cells, and promoted the intestinal permeability to achieve higher oral bioavailability. Especially, we also noticed slightly toxicity for D-MSN on RAW264.7 and DC 2.4 cells. So, we concluded that L-Ala modified NPs were the preferential conformations on account of the high efficiency and low toxicity, and further modified VSN with L-Ala for functional bionics.

Considering the review's advice, we had added a chiral control of DVSN (VSN modified with D-Ala) and supplied a comparison between CVSN and DVSN on some key indicators, including amino acid adsorption, intestinal bio-adhesion, intestinal permeability, and cellular uptake. In accordance with our preliminary results on L/D-MSR and L/D-MSN, CVSN with L-chirality not only showed higher adsorption amount on L-amino acid (17.58 $\mu\text{g}/\text{mg}$) than D-Ala (7.72 $\mu\text{g}/\text{mg}$; Fig. 2k-l), but also showed beneficial intestinal bio-adhesion (single NPs and competitive bio-adhesion between two NPs, Fig. S14), intestinal permeability (Fig. S16) and cellular uptake (Fig. S27), demonstrating that surface chirality dictated mucosal interactions via active chiral recognition.

3) Besides, we had rewritten the synthesis parts with more clarifications, in which AVSN were defined in the text, with more information on the synthesis steps that separate MSNs, VSNs, AVSNs, and CVSNs Please see the sentence marked in blue in Page 24. We had also revised the schematic diagram and structural formula to give more information about the material synthesis (Fig. 1a and Fig. S1), and all experimental procedure and dosage were provided. By the way, a qualitative measurement on the surface coverage proportion of L-Ala was calculated to be ~14.5% according to the TGA and EDS results (the amount of alanine grafted on CVSN was calculated to be 14.75% and 14.30% by TGA and EDS data, respectively).

4) Finally, we had increased the comparison between VSN and CVSN in the text. That is, CVSN showed active surface/interface advantages and exhibited advantage in biological processes, including contact (reduced the contact angle from 19.7° to 14.45°; Fig. 2g-f), recognition (showed distinct amino acid adsorption; Fig. 2k-l), bio-adhesion (increased up to 82.95% in retention amount and 332.90 μm in depth of penetration; Fig. 3), permeation

(increased up to 77.43% in uptake amount and 58.19% in depth of penetration; Fig. 4 and Fig. S17), cellular uptake (up to 1.18 times; Fig. 6 and Fig.S25), adsorption (smaller T_{max} , larger C_{max} and the largest AUC; Fig. 5n), and distribution (higher distribution in lung and brain; Fig. 5l–m and Fig. S24). Also the CVSNS based Nano-DDS (IMC@CVSN) showed obvious superiority in pharmacokinetics and pharmacodynamics compared to VSN (Fig. 7e–p, Fig. S33 and Table S2). Additionally, we also supplied the competitive bio-adhesion study of NPs on the human intestinal mucosa and the competitive adsorption of NPs in the intestinal villi of mice, wherein CVSNS always defeated VSN and showed higher retention and penetration on the small intestine (Fig. S15). Based on these results, we confirmed that the chirality of NPs was functioned while performing the oral delivery task.

3. Throughout the manuscript, it would be helpful to include more descriptive figure captions, and imperative to provide larger labels in figure panels as the current figures are often illegible. There are additionally some supplementary figures where the y axes are not labelled. Finally, some figures use the same color scheme for the set of 4 NP materials, but the identity linked between each color changes between panels – these should be standardized.

Response: Thanks for your nice suggestion. The quality of all images were improved, in which the images, labels, arrows and font size were amplified for assessment of accuracy, the y axes were labelled, more descriptive figure captions were supplied, and the figure numbers were rearranged and presented in consecutive order. Besides, we had standardized the color of 4 NPs, wherein black, red, green and blue represented for SSN, MSN, VSN and CVSNS, respectively.

4. The 3D orientation of Figure 2M makes it difficult for the reader to understand where nanoparticles are localizing. Orthogonal projections to a 2D plane may be better to represent this data. Additionally, the red/green color scheme is not friendly to readers with color blindness, and the authors should consider using a different one.

Response: Many thanks for your kind suggestion. We had supplied the 2D plane orientation pictures of NPs permeating through the mucus, which helped to locate the penetration depth and relative location of NPs. Furthermore, the red/green color scheme was changed to a red/blue color scheme to improve the contrast ratio as well as to be friendly to the readers with color blindness. Please see the revised Fig. 2m. As can be seen from the 3D images of CLSM, only weak fluorescent signals from SSN and MSN were distributed in the mucus on the proximal luminal side of the intestine, whereas potent fluorescent signals from VSN and CVSNS were observed in the entire mucus layer with good extensibility (Fig. 2m–o). Particularly, CVSNS could effectively diffuse into the deepest and widest sections of the mucus layer, evidenced by co-localization of fluorescence signals in the 2D plane orientation pictures.

5. For the adhesion assay outlined in Figure 3, it has generally been difficult for the field to

maintain proper mucus coverage on ex vivo tissue, and the authors do not provide any evidence that theirs is intact. Either characterization of this model, or citations to previous work that did properly characterize it, should be provided to ensure that particles are adhering to biologically relevant tissue and not deprotected epithelial cells. In addition, Figure 3 should be updated to more clearly show which panels correspond to mouse intestines and which to human intestines.

Response: Many thanks for your kind suggestion. During the experimental procedure, we had performed a lot of studies on the intestinal mucosal tissue, including the collection of intestinal mucus and the bio-adhesion of NPs on the mucosal tissue. The former corresponded to the removal of mucus layer, wherein the texture of the epithelium was clear and lack of gloss, and was quite different with the latter that colloidal mucus layer with gloss was covered on the tissues. Especially in the human tissues (which was large in shape), the colloidal mucus layer was clearly visible with physiological yellow color due to the mix of bile at the major duodenal papilla, enabling the fast recognition of mucus layer.

2) Considering your kind suggestion, to study whether proper mucus coverage was maintained on *ex vivo* tissue, we treated the rat and human intestines under two conditions, including the place of tissue at 4°C for 12 h with the exist of Tyrode's solution to simulate the storage of tissues (in fact, the experiment was basically performed within 2 h storage), and the sequential exposure (at R.T.)-incubation (at 37°C)-elution procedure to represent the experimental process. For the two conditions, we used a light microscope to observe the mucosal surface, wherein all mucosa showed intact and proper mucus coverage (Fig. S12). Based on the clinical experience and experimental results, we confirmed the proper mucus coverage on *ex vivo* tissue. Please see the revised Fig.S12.

3) In addition, we had updated the figures and clearly marked which panels corresponded to mouse intestines and which to human intestines. Please see the revised Fig. 3.

6. *For the discussion of Figure 5, the authors should provide more information on how Rhodamine is incorporated into the nanoparticles, as well as how this could affect experimental results. For example, is rhodamine able to leach back out of the particles? If so, the author's conclusions about efflux of SSNs and about partition coefficients both more likely reflect the behavior of free rhodamine. The authors should also discuss, with sources, the implications of their findings of silica accumulating in heart tissue, as well as include information on the time points for histological analysis.*

Response: 1) In the present study, we chose the safe and non-toxic fluorescent dyes RITC and FITC for *in vitro* and *in vivo* tracing, which was incorporated into the NPs through covalent bond via three steps. i) the isocyanic acid groups in RITC and FITC reacted with the amino group in APTES via nucleophilic addition and formed sphenylthiourea compounds (Reaction 1 in Fig.S9). ii) the dye precursor reagents RITC-APTES or FITC-APTES hydrolyzed and condensated with the silica framework of NPs and formed stable chemical bonds (Reaction 2 in

Fig.S9). In this way, the fluorescent dyes were incorporated into the NPs through covalent bond. iii) We then washed and centrifuged the RITC and FITC labeled NPs repeatedly until the supernatant was colorless to remove the unreacted dye precursor reagents.

2) To analyze the stability of fluorescently NPs, RITC labeled NPs were cultured with 3 ml simulated gastric fluid (SGF), simulated intestinal fluid (SIF), and simulated body fluid (SBF) at 37°C, respectively. At the predesigned points (0, 2, 6, 12 and 24 h), RITC labeled NPs suspension were drawn out and analyzed using microplate reader to record the fluorescence spectra. It was indicated that RITC labeled NPs was a robust fluorescent probe with good fluorescence stability (confirmed by the almost constant peak pattern; Fig. S10). Afterwards, the supernatant was collected by centrifugation and recorded the fluorescence spectra, which indicated that no fluorescent dye was released during the incubation periods as evidenced by the extremely low fluorescent signal. In a word, other than the electrostatic adsorption or physical loading of fluorescent dyes onto the surface or nanopores of the NPs, the covalent grafting strategy used in this study confirmed good fluorescent stability, and enabled the *in vivo* tracing of NPs. Moreover, weight loss and SEM were used to investigate the stability of the NPs, wherein all NPs kept the integrity of most particles and basically excluded the leakage of fluorescence dyes caused by the disassembly of NPs. So, we think in the efflux of SSN as well as the partition coefficients were more likely to reflect the behavior of NPs.

3) Besides, NPs distribution in the major organs was qualitatively and quantitatively analyzed by IVIS Lumina Series III Living Image System and ICP-MS, respectively. As shown in Fig. S24, the fluorescence signal of CVSN was highest in almost all organs, implying satisfactory bioavailability, while SSN and MSN were poorly distributed in organs. To be specific, in lung and brain, with abundant vascularity and numerous capillaries, the fluorescence intensity and Si concentration ranked as CVSN > VSN > MSN > SSN at 24 h and 7 day post administration (Fig. 5l-m). However, VSN were highly distributed in heart, spleen and liver at all time points, wherein the macrophage was rich, due to the passive targeting of NPs. It should be noticed that, the liver was still the largest passive targeted organ for all the NPs owing to its large weight. Collectively, NPs exhibiting different surface topology showed distinguished distribution trends and biological fate, and CVSN exhibited the best oral adsorption and distribution characteristics.

7. *The mechanism presented in Figure 6A is done so without evidence, and is never directly supported in the text. The authors must properly justify these claims. It should also be noted that Supplementary Figure Panels 22C through 22G are never called from the main text.*

Response: We sincerely appreciate the valuable comments. In the experimental conception phase, we designed NPs exhibiting virus-mimic surface topology and aimed to explore its structure–function relationships on biological effect by comparing with NPs exhibiting different surface roughness (including the smooth SSN, mesoporous MSN and virus-like VSN). In our

assumption, spherical SSN had smooth surface, which was associated with good mobility and small friction coefficient, and was hard to provide a fixed friction surface. For MSN, VSN and CVSN, the surface roughness increased in turn. The pore structure as well as the spiky surface was able to provide definite contact points, providing material basics for the sliding friction. Once contacting with the biological membranes, MSN, VSN and CVSN was theoretically related to the large frictional force comparing to SSN. However, due to technological limitation, the assumption had not been directly verified, although many studies indicated the improved adhesion and penetration effect of CVSN both *in vitro* and *in vivo*. As a validation, we directly observed the internalization and transport of NPs via bio-TEM. As expect, spherical NPs (SSN and MSN) showed poor adhesion on the cell membrane with single contact point and followed by the limited endocytosis into the cells, while VSN and CVSN with virus-mimic surface topology provided multiple contact sites and directly anchored on the cell membrane to facilitate cellular uptake (Fig. 6a). Excitingly, numerous CVSN appeared in the cells. Considering your advice, we had deleted the relative descriptions about rolling friction and sliding fraction after carefully consideration, and remained the descriptions on “single-site contact” and “multi-site contact”, which was directly observed by bio-TEM. Once again, thanks for your constructive suggestions.

In addition, supplementary figure panels 22C-G (revised to Fig. S29), relevant to the dynamic body weight, the organ/body ratios, the representative histopathological images of the major organs, the hematology results and biochemical results of mice after oral treatment of NPs, had been added in the manuscript. To detect the underlying *in vivo* toxicity of NPs, mice received SSN, MSN, VSN or CVSN exposure were executed on the 14th day, and their blood and major organs were collected and subjected to hematological and biochemical tests, and histopathological examinations, respectively (Fig. S29). No sudden death, unusual behaviors, and significant weight loss were observed during the experimental period. The organ/body ratios (%) of the major tissues (heart, liver, spleen, lung and kidney) appeared to be normal. H&E staining images revealed that SSN, MSN, VSN and CVSN would not cause obvious histopathological abnormalities or damage after oral administration. Furthermore, the hematological and biochemical parameters were all within the reference ranges and showed no substantial difference between the groups. In a word, the NPs lacked any indication of toxicity both *in vitro* and *in vivo*. Please see the relevant expression marked in blue in Page 17.

8. *The authors mention poor solubility of indomethacin, but otherwise do not justify its use as a model drug, which raises concerns given the 100% bioavailability of orally administered indomethacin (<https://pubmed.ncbi.nlm.nih.gov/26865183/>). The drug release and particle uptake data presented better support the idea that the Nano-DDS developed here are releasing indomethacin into the intestinal lumen, and that the free small molecule is diffusing on its own across the epithelial barrier. To this end, the authors should consider splitting the massive amount of data that they have generated into two separate manuscripts: (1) using MSNs or*

VSNs to deliver epithelium-permeable drugs like indomethacin into the intestines for rapid adsorption, and (2) chiral functionalization of VSNs to CVSNs to drive nanoparticle uptake across the epithelium and deliver drugs that are not able to cross the barrier on their own.

Response: Many thanks for your nice suggestion. **1)** To our knowledge, most drugs are suffered from unsatisfied oral bioavailability as a consequence of the poor solubility and/or low permeability, and that is the foundation for the BCS classification of drug. Meanwhile, the harsh *in vitro* and *in vivo* environment, especially the GIT conditions, also challenged the stability of drug. Faced such dilemma, we aimed to construct an efficient oral Nano-DDS to overcome the stability, solubility and permeability bottlenecks of drugs by simulating the structure and function of viruses in this study. To be specific, IMC was loaded into the NPs to simulate the package of genes, wherein the payloads were bound into the nanopores and high dispersed in amorphous state to isolated from the biological environment and to improve their stability and solubility, while the chiral nanospikes multi-sited anchored and chiral recognized on the intestinal mucosa to enhance the penetrability. By mimicking the shape, size, surface topology, chiral recognition and gene encapsulation of viruses, the payload IMC achieve favorable oral delivery efficiency.

2) However, as the reviewer pointed out, IMC released from the CVSNS without controlled manner. This was mainly because drugs could be incorporated into CVSNS with high efficiency along with the change of crystalline state to amorphous form (as confirmed by the XRD and DSC results). Then drug molecules could be easily dissolved out, and resulted in the enhancement on both the release rate and amount. As demonstrated, CVSNS exhibited superior bio-adhesion and intestinal permeability, which then enhanced the opportunities for the oral adsorption of drug carriers and the guest payloads. Even if the drug molecules were released, CVSNS still helped them to remain on the mucosa for a prolonged period and resulted to a passive absorption. As confirmed by the pharmacokinetics and pharmacodynamics studies, IMC@CVSNS achieved favorable oral drug delivery efficiency (relative bioavailability 1145.9%; Fig. 7e and Table S2), and displayed good anti-inflammatory effects (inhibition rate: 241.01%–2315.86%; Fig.7 j and n, and Table S3–4). The greatly improvement on the oral absorption was attribute to both the increased drug dissolution and the favorable performance of CVSNS in the GI tract which carried the drug in the nanopores passing through the multiple physiological barriers and helped the passive uptake of the released drug.

In addition, we had researched extensive literatures and found the absolute bioavailability of IMC between 7%–35%. Also, we used to apply IMC as model drug for oral delivery in our preliminary work, and noticed a similar pharmacokinetic behavior of pure IMC with this time. To further confirm the improved oral bioavailability of IMC by using CVSNS as Nano-DDS, we also investigated the *ex vivo* intestinal transport via everted intestinal sacs model, and the results was coordinated to the pharmacokinetic data, wherein the transport amount of IMC@CVSNS was up to 7.70-times higher than that of IMC (Fig. S31).

3) More importantly, we carefully studied your professional suggestions, and found the single verification of IMC was insufficient owing to its fast release manner. After deliberate consideration, we decided to add a versatility verification of CVS N as oral Nano-DDS, wherein drugs with diverse structures, molecular weights, functional groups, solubility and permeability were involved. We selected a series of NSAIDs with problems of oral adsorption (including nimesulide [NMS], acetaminophen [AC], aspirin [ASP], celecoxib [CEL], flurbiprofen [FB], ibuprofen [IBU] and IMC, with similar pharmacodynamic effects for easier comparison) belonging to BCS 2–4, characterized by poor solubility and/or low permeability (not able to cross the barrier on their own; Fig. 8b and Table S5). We encapsulated them into CVS N to construct Nano-DDS, confirmed the successfully loading (Fig. 8d and Fig. S30) and high dispersion of drugs in amorphous state via FTIR and XRD (Fig. S34), and investigated the *ex vivo* intestinal transport and *in vivo* bioavailability of these Nano-DDS via everted intestinal sacs model (Fig. 8a) and MEST model (Fig. 8m), respectively. The results indicated that CVS N was able to improve the oral adsorption of almost all these NSAIDs, wherein the transport rate raised about 1.60–7.71 times (Fig. 8c and e–i) and the anti-inflammatory effect improved about 1.21–2.29 time (Fig. 8n–p and Table S6). By this mean, we demonstrated CVS N was an efficient oral Nano-DDS to overcome the stability, solubility and permeability bottlenecks of drugs, and hope the change would meet the approval.

9. *For the discussion of Figure 8, the authors should consider adding a pathologists' report with quantitative metrics to support the conclusions that they draw from histology images. The arguments would be better supported with some literature citations justifying pre-treatment of animals in the inflammation models, as well as establishing the two models employed. The figure itself would also benefit from better delineation between panels – which correspond to ankle swelling vs. ear?*

Response: Many thanks for your advice. The anti-inflammatory biological effects of the IMC loaded NPs were estimated by mouse ankle swelling test (MAST), mouse ear swelling test (MEST) and mouse writhing test (MWT). By direct measurement on the ankle circumference, ear pieces weight and writhing numbers of experimental animals, we noticed the relief on swelling in the IMC loaded NPs treated groups, wherein IMC@CVSN always showed the best therapeutic effect. In details, the swelling rates on MAST were 16.41%, 11.18%, 6.76%, 2.77% and 2.49% for animals in the saline, IMC, IMC@MSN, IMC@VSN and IMC@CVSN groups, respectively (Fig. 7j). The swelling rate on MEST were 90.96%, 53.77%, 36.06%, 29.98% and 21.79%, for animals in the saline, IMC, IMC@MSN, IMC@VSN and IMC@CVSN groups, respectively (Fig. 7n). The writhing numbers on MWT were 27.00, 18.33, 10.33, 6.67 and 7.00 for animals in the saline, IMC, IMC@MSN, IMC@VSN and IMC@CVSN groups, respectively (Fig. S33). For CVS N, the inhibition rates were measured to be 241.01%–2315.86%, 153.73% and 230.77% in MAST, MEST and MWT, respectively (Table S3–4). From a macroscopic

perspective, we found IMC@CVSN exhibited highest oral bioavailability and showed the best therapeutic effect. According to your advice, we had supported with some literature citations for the pre-treatment of animals in the models.

Then we sent the relevant swelling tissues for histopathological examination (H&E staining) to appraisal the microscopic damage. For your advice on quantitative metrics, we had labeled the inflammation-related indicators on the section (wherein the numbers of the correlated indexes could be directly observed from the number and colors of arrows), added the disease damage score (the damage score was appraised according the severity of tissue injury on a 0–4 scale: 0–almost normal tissue, 1–one or two lesions, 2–severe lesions, 3–very severe lesions, and 4–tissue full of lesions) with the assistance of a pathologist, and updated the images and descriptions in the manuscript (Fig. 7l and p). Conformance to the macroscopic findings, IMC@CVSN also displayed the best anti-inflammatory effect in the microscopic scope, with the slightest edematous and the smallest numbers of inflammatory cells in both MAST and MEST. And the disease damage score decreased from 3.83 to 0.83, and from 2.17 to 0.67 in MAST and MEST, respectively (Fig. 7k and o). We had rechecked with the pathologist, referred to relevant literatures and revised the histopathological examination part, in which the images, labels, arrows and font size were amplified for assessment of accuracy, more descriptive figure captions were supplied, and the figure numbers were rearranged and presented in consecutive order. Please see the sentences marked in blue in Page 20–21.

In one word, based on the efficacy-time curves, the swelling and inhibition rates, the representative physical appearances, the histopathological images and the disease damage score, we verified the best therapeutic effect of IMC@CVSN.

10. The authors should make sure that their conclusions do not overreach the work shown. In line 520, it is unclear from the manuscript what is meant by “signal transfer at the bio-interface.” Similarly, line 532 mentions “low cost, high stability, ease of large-scale production” that are not discussed in the main text.

Response: Many thanks for your kind work. The language was carefully revised to ensure clarity of the point made and conclusions, with the guiding principle of “the conclusions do not overreach the work shown”. The unvalidated contents, such as “signal transfer at the bio-interface.”, “low cost” and “ease of large-scale production” were deleted from the main text. We appreciated for reviewers’ warm work earnestly, and hope that the correction would meet with approval. Once again, thanks for your kind work.

Yours Sincerely,

Heran Li
China Medical University

110122 Shenyang

PR China

E-mail: liheranmm@163.com

Reviewers' Comments:

Reviewer #1:

Remarks to the Author:

The authors have carefully revised the manuscript according to the reviewers' comments. I agree to publish it on Nature Communications.

Reviewer #2:

Remarks to the Author:

The authors have thoroughly addressed the concerns raised in primary review. A few small points remain:

- 1) Some figure panels and axis labels – especially in Supplementary Figure 29 – are still too small to read, and become blurry upon zooming in.
- 2) In the new Figure 8b, Acetaminophen is listed as both Class III and Class IV

REVIEWERS' COMMENTS

Reviewer #1 (Remarks to the Author):

The authors have carefully revised the manuscript according to the reviewers' comments. I agree to publish it on Nature Communications.

Response: We sincerely appreciate your positive comments and professional review work on our article.

Reviewer #2 (Remarks to the Author):

The authors have thoroughly addressed the concerns raised in primary review. A few small points remain:

- 1) Some figure panels and axis labels – especially in Supplementary Figure 29 – are still too small to read, and become blurry upon zooming in.
- 2) In the new Figure 8b, Acetaminophen is listed as both Class III and Class IV

Response: We sincerely appreciate your positive comments and professional review work on our article. 1) All panels and axis labels were amplified, especially in Supplementary Figure 29 (revised to Supplementary Figure 30). Please see the revised Supplementary Figure 30.

2) In the new Figure 8b, Acetaminophen was listed as Class IV. Please see the revised Figure 8b.